# Understanding day-night differences in dust aerosols over the dust belt of North Africa, the Middle East, and Asia

Jacob Zora-Oni Tindan*, Qinjian Jin, Bing Pu

Department of Geography and Atmospheric Science, University of Kansas, Lawrence, KS, USA

*Correspondence to*: Jacob Zora-Oni Tindan (jztindan@psu.edu), Bing Pu (bpu@ku.edu)

* Now at Department of Meteorology and Atmospheric Science, Pennsylvanian State University, State College, PA, USA

**Abstract.** Utilizing the well-calibrated, high spectral resolution, and equal-quality-performance of daytime and nighttime observations (9:30 a.m. and 9:30 p.m. local solar equator-crossing time (local solar ECT)) products of the Infrared Atmospheric Sounder Interferometer (IASI) from the Laboratoire de Météorologie Dynamique (LMD), this study investigates the day-night

differences in dust aerosols over the global dust belt of North Africa, the Middle East, and Asia. Both daytime and nighttime dust optical depth (DOD) at 10 µm shows high consistency with solar and lunar observations of coarse mode aerosol optical depth (CAOD) from AErosol RObotic NETwork (AERONET) sites across the dust belt, with correlation coefficients of 0.8–0.9 for most sites. Both IASI DOD and dust layer height show significant (95% confidence level) day-night differences in dust aerosols over the major dust sources within the dust belt. Daytime DOD over the central to the northern Sahara Desert, the

central to eastern Arabian Peninsula, and the Taklamakan Desert is significantly higher than that of nighttime, but lower than nighttime over the southern Sahel to the Guinea Coast, and the western to central Indian subcontinent in the annual mean. The magnitude of the day-night differences in DOD is larger and evident in boreal winter and spring than other seasons. The positive day-night differences in DOD (i.e., higher daytime values than nighttime) over the central Sahara, the Middle East, and Asia are likely associated with greater dust emissions driven by higher dust uplift potential (DUP) and stronger wind speed at

daytime. Dust layer heights demonstrate negative day-night differences over dust source regions in the central Sahara, central Arabian Peninsula, and the Taklamakan Desert, and positive height differences in the southern Sahel to the Guinea Coast, southern parts of the Arabian Peninsula, and large parts of the Indian subcontinent. The higher dust layer height over the Guinea Coast and the Indian subcontinent during the daytime is associated with a deeper planetary boundary layer height and greater convective instability at daytime than nighttime, which promotes vertical transport and mixing of dust aerosols. The

corresponding lower daytime DOD over the Sahel and the Indian Subcontinent indicates a possible dilution of dust aerosols when they are transported to higher altitude by convections where they are more susceptible to horizontal transport.

Ground-based observations of dust show surface $PM_{10}$ concentration and CAOD exhibit a spatially varying diurnal cycle across the dust belt. CAOD and $PM_{10}$ concentrations peak in late morning and from late afternoon to midnight in the Sahel, early afternoon and around midnight in the Middle East, the timings of which are largely consistent with the day-night

differences in IASI DOD. It is also found that DOD from reanalysis products (e.g., Modern-Era Retrospective Analysis for Research and Applications, version 2 (MERRA-2) and ECMWF Atmospheric Composition Reanalysis 4 (EAC4)) failed to

capture the day-night differences in IASI DOD in large parts of the dust belt except in small dust source hotspots over North Africa.

## 1 Introduction

Mineral dust is one of the primary aerosol species in the atmosphere and forms an integral part of the climate system. It is produced by wind erosion in deserts, dry lake beds, arid and semi-arid regions (Penner et al., 2001). The uplift of dust aerosols over source regions mostly occurs when the surface wind speed, which is also affected by land surface characteristics and vegetative cover, exceeds a suitable threshold (Fernandez-Partagas et al., 1986; Marsham et al., 2008; Bergametti et al., 2017; Pu et al., 2020). The global emission of dust aerosols is estimated to range between 1000 and 5000 Tg yr$^{-1}$ with high

spatiotemporal variability (Duce, 1995; Ginoux et al., 2001; Huneeus et al., 2011; Checa-Garcia et al., 2021). North Africa alone accounts for about 50% of the global dust emissions (Schütz, 1980; D'Almeida, 1986; Tegen and Fung, 1994; Swap et al., 1996; Ginoux et al., 2012; Kok et al., 2021), followed by the Middle East and Asia contributing about 40% of global dust emissions (Prospero et al., 2002; Goudie and Middleton, 2006; Tanaka and Chiba, 2006; Huneeus et al., 2011; Kok et al., 2021).

Dust aerosol impacts atmospheric radiative balance directly by dust–radiation interactions and indirectly by dust–cloud interactions, with the latter being one of the largest sources of uncertainties in modelling aerosol effects in global climate change (Forster et al., 2007; Haywood et al., 2005; Mahowald et al., 2010; Yan et al., 2015; Adebiyi and Kok, 2020). The radiative effect of dust refers to its scattering and absorption of incoming shortwave and outgoing longwave radiation as well as thermal infrared emissions, consequently affecting regional climate, e.g., African and Indian monsoon systems (Miller and

Tegen, 1998; Li et al., 2004; Mahowald et al., 2010; Jin et al., 2014, 2021) and tropical cyclones in the North Atlantic (Karyampudi and Carlson, 1988; Dunion and Velden, 2004; Wong and Dessler, 2005; Strong et al., 2018). Dust aerosols can also modify the macro– and micro–physical properties of clouds by serving as cloud condensation and ice nuclei, namely aerosol–cloud interactions that can further interact with the hydrological cycle (Levin et al., 1996; Rosenfield et al., 1997; Nakajima et al., 2001; DeMott et al., 2003; Bangert et al., 2012). When dust aerosols are deposited into the ocean and land,

they provide nutrients such as phosphorus, iron, and nitrogen to continental and maritime ecosystems (Duce and Tindale, 1991; Mills et al., 2004; Okin et al., 2004). For instance, African dust has been found to influence ecosystems in the Amazon Basin (Swap et al., 1992; Bristow et al., 2010; Yu et al., 2015) and the Atlantic Ocean (Jickells et al., 2005; Mahowald et al., 2010).

Quantifying the climatic impacts of dust requires accurate and detailed information on their spatial and temporal distributions. In addition to seasonal, interannual, and decadal timescales of variability, the diurnal variation in dust is also an

important aspect that has been explored by many works. Past studies reveal significant daytime and nighttime variabilities in dust loading over the dust belt (Wang et al., 2004; Schepanski et al. 2009; Fiedler et al., 2013; Heinold et al., 2013; Kocha et al., 2013; Osipov et al., 2015; Yu et al., 2016; Chédin et al., 2020; Yu et al., 2021). For example, in North Africa, pronounced dust emissions during morning hours are found to be associated with the breaking down of the nocturnal low–level jets

(Engelstaedter et al., 2006; Todd et al., 2008; Tulet et al., 2010; Knippertz and Todd, 2012) and in the late afternoon period as a result of mesoscale convective systems that generate dust emissions at the leading edge of density currents (Flamant et al., 2007; Marsham et al., 2008; Todd et al., 2008; Knippertz and Todd, 2012). Satellite observations and regional model simulations in West Africa showed a well-marked diurnal variability of dust associated with a rising planetary boundary layer maximizing at about 15 UTC (Coordinated Universal Time; about 4 p.m. local solar time (LST)) (Chaboureau et al., 2007). Using the fifteen-minute Meteosat Second Generation (MSG) Spinning Enhanced Visible and Infrared Imager (SEVIRI) satellite product, Schepanski et al. (2009) found about 65% of the dust source activation in the Sahara Desert occurring between 0600 and 0900 UTC (about 5:00 a.m. –10:00 a.m. local solar time at the western and eastern boundaries of the Sahara).

In the Middle East, summertime dust emissions are primarily caused by the strong, persistent Shamal winds which maximize around local noon over the Iraqi Desert (Yu et al., 2016). Around the Gobi and Taklamakan deserts in Asia, dust emissions in spring to early summer show a diurnal change of more than ±10% Aerosol Optical Depth (AOD) and ±30% of Angström exponent, with larger AOD and smaller Angström exponent values in late afternoon (Wang et al., 2004). Smirnov et al. (2002) showed an increase of AOD by 10%–40% during the daytime over dust sources in North Africa and Asia with less diurnal variability over regions where dust aerosol is a major contributor to the total AOD. By analysing aerosol extinction and typing profiles from Cloud-Aerosol Transport System (CATS) lidar on a global scale, Yu et al. (2021) identified a significant daytime and nighttime variations in dust and dust mixture loading over the major dust sources in North Africa, and western and southern North America.

However, observations of the full diurnal cycle of dust with a global coverage is still lacking. Ground-based instruments such as AErosol RObotic NETwork (AERONET; Holben et al., 1998; O'Neill et al., 2003) and Laboratoire Interuniversitaire des Systèmes Atmosphériques (LISA) stations over the Sahel (Marticorena et al., 2010) have high temporal resolution (~5–15 minutes for AERONET and hourly for LISA), but with low spatial coverage. On the other hand, while satellite products have much higher spatial resolutions and coverage, polar-orbiting instruments have low temporal coverage, i.e., two times daily observations. Moreover, most of these instruments (both satellite and ground-based) sample dust aerosols based on the measurement of radiance in visible bands, making it difficult to observe dust events in the nighttime and thereby missing out some important characteristics of dust. For instance, widely used products, such as, the Moderate Resolution Imaging Spectroradiometer (MODIS) onboard both the Terra and Aqua satellites and Multi-angle Imaging SpectroRadiometer (MISR; Diner et al., 1998) onboard the Terra satellite retrieve AOD once per day only in visible wavelengths. Observations from lidar instruments such as Cloud-Aerosol Lidar with Orthogonal Polarization (CALIOP; Winker et al., 2009) provide vertically resolved aerosol extinction and clouds for snapshots during both daytime (1:30 p.m. local solar equator-crossing time; ECT) and nighttime (1:30 a.m. local solar ECT). However, CALIOP has two significant drawbacks when it is used to study day-night differences in dust optical depth (DOD): (1) A lower signal-to-noise ratio during the daytime than nighttime, making it less sensitive to daytime observations (Liu et al., 2009) and less reliable to directly compare its daytime and nighttime products and (2) A narrow horizontal swath of 5 km and a 16-day repeat cycle, which means there is only one daily observation (afternoon or night) at a specific location thus no day-night differences of DOD can be retrieved at daily timescale. SEVIRI

instrument (Schmetz et al., 2002; Schepanski et al., 2007, 2009) aboard the Meteosat Second Generation satellite, which is a geostationary satellite located at 3.5°W above the equator, provides dust observations from infrared (IR) channels every 15 minutes. However, this product mainly covers Africa and the Arabian Peninsula. The above challenges are partly addressed by the Infrared Atmospheric Sounder Interferometer (IASI; Chalon et al., 2001; Blumstein et al., 2004).

IASI sensor onboard the European Meteorological Operational satellite (MetOP) provides retrievals of dust optical depth (DOD) and dust layer height at IR bands twice per day (9:30 a.m. and 9:30 p.m. local solar ECT) at global scale (Chalon et al., 2001; Klüser et al., 2013; Peyridieu et al., 2013; Capelle et al., 2014, 2018), facilitating the study of day-night variations in dust aerosols. Additionally, coarse mode dust aerosols (CAOD e.g., radius > 1 μm) are more sensitive to infrared (IR) radiation than visible due to their large particle size, so are preferentially retrieved in IR bands (Capelle et al., 2018). IASI has a fine spectral and spatial resolutions of 0.5 cm$^{-1}$ and 12 km at nadir, respectively, as well as showing high quality in capturing the spatiotemporal variability in dust (Hewison et al., 2013) in comparison to ground measurements from AERONET (Capelle et al., 2014, 2018). The observation time of IASI generally coincides with the two dominant dust generation mechanisms in north Africa, the breaking down of the nocturnal low–level jets in the early morning hours and mesoscale convective systems in the late afternoon and early evening period (Engelstaedter et al., 2006; Washington et al., 2006; Knippertz and Todd, 2012; Chédin et al., 2020). One important advantage of IASI is its equal quality performance for daytime and nighttime observations (Hewison et al., 2013; Chédin et al. 2020), making it suitable to compare daytime and nighttime variability of dust. The data have been used to study characteristics of dust in the Sahara Desert (Chédin et al., 2018; 2020).

In this work, we are using IASI DOD and dust layer height products from Laboratoire de Météorologie Dynamique (LMD; Capelle et al., 2018) together with ground-based observations from AERONET and LISA sites (Berkoff et al., 2011; Holben et al., 1998; Marticorena et al., 2010) to understand the daytime and nighttime variability in dust aerosols over the dust belt of North Africa, the Middle East, and East Asia (Fig. 1). Aerosol reanalysis products, such as Modern-Era Retrospective Analysis for Research and Applications (MERRA-2 ; Gelaro et al., 2017; Randles et al., 2017) and ECMWF Atmospheric Composition Reanalysis 4 (EAC4; Inness et al., 2019), which are widely used in model validation and case studies (Grandey et al., 2013; Carmona et al., 2020; Isaza et al., 2021) as they assimilate total AOD from satellite products while providing high spatial and temporal coverage of dust distribution, are employed for comparative purpose with IASI results. We will examine whether these aerosol reanalysis products capture the day-night variations in dust shown in satellite products. Lastly, we will examine the meteorological conditions that contribute to the observed day-night variabilities in dust aerosols. Section 2 describes the study domain and introduces the datasets and data analysis techniques. Results are presented in section 3, and uncertainties are discussed in section 4. Major findings are summarized in section 5.

## 2 Data and methodology

### 2.1 Study Domain

In this paper, we focus on the dust belt extending from North Africa through the Middle East and Central Asia to the deserts in western East Asia (Fig. 1). The Saharan dust belt (0–35°N, 16°W–25°E) is the world's largest source of aeolian desert dust aerosols, with an annual emissions of 400–700 × 10⁶ tons of dust aerosols (D'Almeida, 1986; Schütz, 1980; Swap et al., 1996). There are two major dust sources within the Sahara — the Bodélé Depression in Chad and an area covering eastern Mauritania, western Mali and southern Algeria (Middleton and Goudie, 2001; Schepanski et al., 2007; Ginoux et al., 2012; Yu et al., 2018).

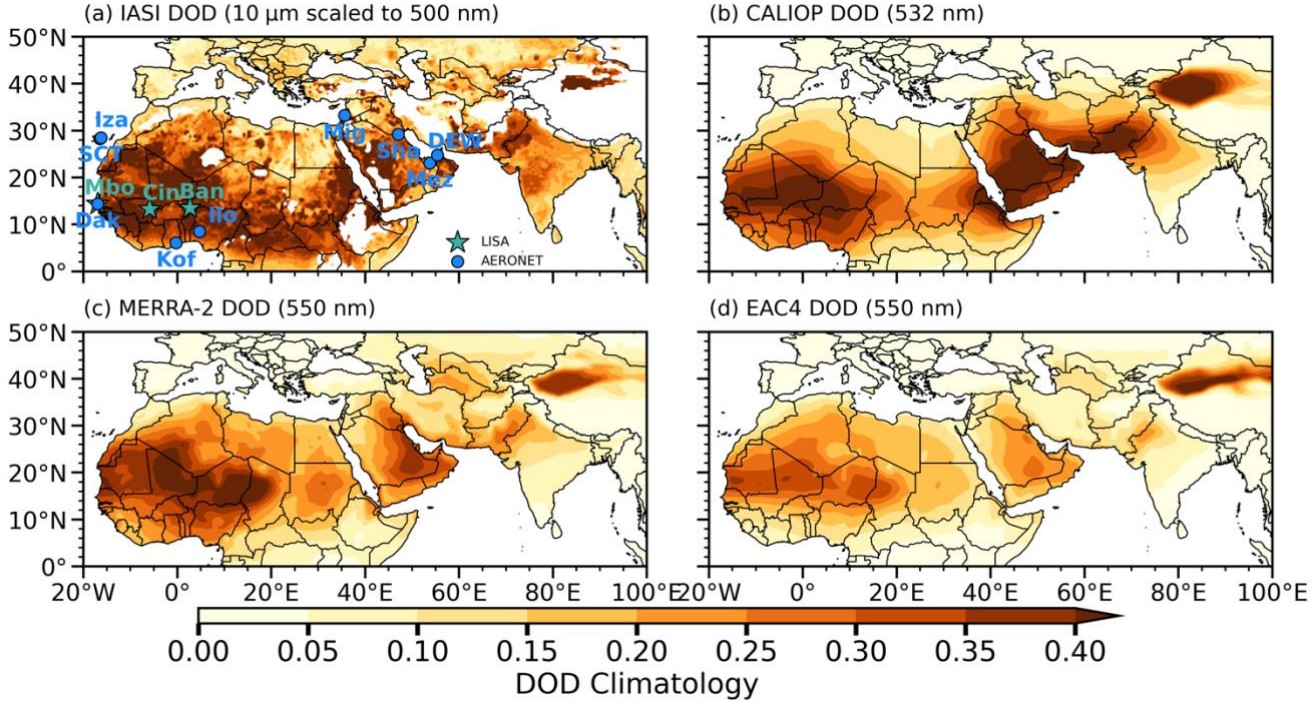

Figure 1: Climatology (2008–2020) of DOD from (a) IASI (LMD version), (b) CALIOP, (c) MERRA-2 and (d) EAC4. Blue-dots denote AERONET sites with both solar and lunar data that are used to examine day-night differences in CAOD. Cyan stars represent LISA sites. Note that IASI DOD in (a) represents the climatology of average daytime and nighttime DOD and is scaled from 10 μm infrared (IR) to 500 nm visible wavelength (VIS) using an IR/VIS ratio of 0.60 (see text for details)). Note that CALIOP data is up to July 2020.

The Middle East dust belt (13°N–38°N, 25°E–60°E) is the world's second largest dust source (Prospero et al., 2002; Goudie and Middleton, 2006; Huneeus et al., 2011; Kok et al., 2021). The Middle East and South Asian dust sources covers Sudan, the Arabian Peninsula, parts of Iran and Afghanistan, and Pakistan (Rezazadeh et al., 2013; Ginoux et al., 2012; Yu et al., 2018). The East Asian dust belt (7°N–46°N, 60°E–100°E) mainly includes the Taklamakan and the Gobi Deserts (Prospero

et al., 2002; Zhang et al., 2003; Ginoux et al., 2012) and is estimated to account for about 3–11% of global dust emissions
(Tanaka and Chiba, 2006). We did not include the Gobi Desert in our domain due to large area of missing data in IASI.

## 2.2 Datasets

This study mainly uses the 10 μm DOD retrieved from IASI (LMD version) as the primary dataset to understand the day-night differences in DOD and dust plume layer height, in the dust belt, along with ground-based observations. Results from IASI are compared with aerosol products from reanalyses. Meteorological variables from reanalysis and stations are used to examine
their influences on the day-night differences in dust aerosols. All the datasets used in this study are summarized in Table 1.

Table 1. Summary of datasets and variables used in this study

| Variable | Dataset | Version | Period used | Spatial Resolution | Temporal Resolution | Link to data |
|---|---|---|---|---|---|---|
| **DOD, dust layer height** | IASI | LMD v2.20 | 2008-2020 | 12km | ~12 hourly | https://iasi.aeris-data.fr/DUST-AOD_IASI_A_data/ |
| **DOD, dust layer height** | CALIOP | 4.20 | 2008-2020 | 5 km (5°×2°) | monthly | https://asdc.larc.nasa.gov/project/CALIPSO |
| **CAOD** | AERONET | 3.0 | 2008-2020 | station | 5–15 mins | https://aeronet.gsfc.nasa.gov/ |
| **PM$_{10}$** | LISA | - | 2008-2020 | station | hourly | http://www.lisa.u-pec.fr/SDT/index.php?p=3 |
| **DOD** | MERRA-2 | - | 2008-2020 | 0.625°×0.5° | hourly | https://disc.gsfc.nasa.gov/ |
| **DOD** | EAC4 | V4 | 2008-2020 | 80 km | 3-hourly | https://ads.atmosphere.copernicus.eu |
| **Rainfall, PBLH, CAPE, circulations** | ERA5 | - | 2008-2020 | 0.25°×0.25° | hourly | https://cds.climate.copernicus.eu/#!/home |
| **Precipitation** | IMERG | V06B | 2008-2020 | 0.1°×0.1° | 30 mins | https://gpm.nasa.gov/data/directory |

## 2.2.1 LMD IASI

IASI is a high spectral resolution thermal infrared Fourier transform spectrometer (Chalon et al., 2001; Blumstein et al., 2004) onboard MetOP-A, MetOP-B and MetOP-C satellites. It measures radiance over 8641 spectral channels extending from 645 to 2760 cm$^{-1}$ with a spectral resolution of 0.5 cm$^{-1}$ after apodization. It has a ground resolution of 12 km at nadir. Onboard MetOP-A at an altitude of about 800 km, IASI observes Earth at an angle of up to 48.5° perpendicular to both sides of the satellite track. This corresponds to a swath width of ~2,200 km leading to an approximate global coverage in 12 hours.
The satellite has a local solar equator-crossing time (local solar ECT) of approximately 9:30 a.m. and 9:30 p.m. and is available from July 2007 to October 2021 as at January 2023. MetOP-B was launched in September 2012 and has been operational since February 2013 while MetOP-C was launched at the end of 2017 and has been providing data since 2018. With MetOP-A coming to an end, MetOP-B and -C will continue providing data. The three IASI instruments are expected to provide continuous measurements up to a total of 15 years.
Due to its wide spectrum in longwave range and fine spectral resolution, IASI is widely used to retrieve atmospheric compositions (Clerbaux et al., 2009; Bauduin et al., 2016) during both day and night times. While several retrieval algorithms

are available for IASI DOD and dust layer height (eg., Callewaert et al., 2019; e.g., Clarisse et al., 2019), we use the retrieval from Laboratoire de Météorologie Dynamique (LMD; Peyridieu et al., 2013; Capelle et al., 2014; 2018) as it provides global retrievals of both DOD and dust layer height. IASI dust products show good consistency with ground observations (Capelle et al., 2014, 2018; Peyridieu et al., 2013; Zheng et al., 2022) and good performance in comparison with other IASI DOD datasets (Klüser et al., 2016). LMD IASI has already been used to study characteristics of dust in the Sahara Desert (Chédin et al., 2020). The publicly available L2 data also allow us to validate and compare with ground observations in our study domain, and to interpolate the data to a reasonably high spatial resolution (i.e., 0.5°×0.5°) to facilitate our study. LMD IASI dusty and cloudy pixels are distinguished using cloud mask based on nine screening tests consisting of infrared observations from both IASI and the Advanced Microwave Sounding Unit (AMSU) at same time and locations over the globe (Capelle et al., 2018). The retrieval of DOD and dust layer height from IASI cloud-free observations is based on an iterative two-step approach using different look-up tables (Capelle et al., 2018; Peyridieu et al., 2013). The first step determines the atmospheric state using 18 IASI channels, and the second step is the retrieval of 10 μm DOD, dust layer mean altitude, and surface temperature simultaneously using the algorithm similar to that was originally applied to Atmospheric Infrared Sounder (AIRS; Peyridieu et al., 2010). Here, level 2 (version 2.2.0) daily 10 μm DOD and dust layer height at 9:30 a.m. and 9:30 p.m. local solar ECT (hereafter referred to as daytime and nighttime, respectively) are used and regridded into a 0.5° by 0.5° grid from January 2008 to December 2020. Dust layer height in the dataset is defined as the height at which half of the DOD is found above and the other half below (Peyridieu et al., 2013; Capelle et al., 2018; Chédin et al., 2020).

### 2.2.2 CALIOP

CALIOP is a spaceborne two-wavelength polarization lidar onboard Cloud-Aerosol Lidar and Infrared Pathfinder Satellite Observation (CALIPSO) satellite. It provides high resolution vertical profiles of global clouds and aerosols measurements since June 2006 (Winker et al., 2009). CALIOP is a nadir viewing instrument which has a very narrow swath width i.e., a beam diameter of 70 m at the Earth's surface corresponding to a 16-day repetition cycle with an instantaneous field of view approximately 300 m and 70 m. Level 3 cloud-free monthly DOD at 532 nm and dust layer height on a 5° × 2° grid from 2008 to 2020 are used to compare with IASI. Note that because of the high altitude and modest power-aperture of CALIOP, its daytime product has an extremely low signal-to-noise ratio (Winker et al., 2017), making a direct comparison between daytime and nighttime products less reliable. Moreover, due to its narrow swath width, no day-night difference can be calculated at daily timescale. To compare with IASI dust layer height, we analyzed dust altitude from CALIOP by calculating the mean of the highest and lowest dust aerosol layer detected.

### 2.2.3 AERONET

AERONET is a ground-based sun photometer aerosol observation network established by the National Aeronautics and Space Administration (NASA) and PHOtométrie pour le Traitement Opérationnel de Normalisation Satellitaire

(PHOTONS) which measures atmospheric aerosol properties globally (Holben et al., 1998). The sun photometers perform measurements of solar irradiance in eight spectral bands (340, 380, 440, 500, 670, 870, 940 and 1020 nm) with a field of view 1.2° in about every 5 to15 minutes from 5 a.m. ~ 6 a.m. to 5 p.m. ~ 6 p.m. in Coordinated Universal Time (UTC depending on the site). The lunar photometers perform nocturnal measurements from 5 p.m. or 6 p.m. to 5 a.m. or 6 a.m. UTC with an approximate field of view of 1.29° at eight nominal wavelengths of 440, 500, 675, 870, 937, 938, 1020, and 1640 nm. The estimated uncertainty of AOD from direct solar radiation measurement is about 0.010 ~ 0.021 (wavelength dependent) (Eck et al., 1999).

We use version 3 level 2 (cloud screened and quality assured) Spectral Deconvolution Algorithm (SDA; O'Neill et al., 2003) retrieval of the coarse mode AOD (CAOD; Eck et al., 2010) around 500 nm to approximate DOD and compare with IASI DOD. It is important to note that the SDA algorithm of AERONET CAOD is sensitive to the presence of high clouds such as cirrus and may lead to overestimation of AERONET CAOD (Smirnov et al., 2018). Over coastal regions, CAOD may contain information from sea salt as well, with an estimated contribution of 0.05–0.10 (Spada et al., 2013; Clarisse et al., 2019). Level 2.0 data are not available at lunar sites, so level 1.5 data (cloud screened but not quality assured) are used.

For accurate comparison with IASI, several filtering steps are used to select the AERONET sites as shown in Figs. 1, S1 and Tables 2–3. Firstly, only sites with sample size greater than three years within the dust belt between 20°W–100°E and 0°N–36°N are selected. Secondly, following Capelle et al., (2018), SDA sites with higher root mean square error (RMSE) in CAOD (i.e., RMSE > 0.05 + 0.15 ×CAOD) are removed. This criterion removed a few East Asian sites with low CAOD and high RMSEs. Validation of IASI daytime and nighttime DOD against AERONET solar and lunar retrievals are conducted at 46 sites for daytime, and 11 sites for nighttime (Fig. 1 and Tables 2 and 3). The day-night difference analysis is carried out using sites with both solar and lunar data available on the same days after the filtering processes. Nine sites are found (blue dots in Fig. 1).

### 2.2.4 LISA

A network of three ground-based observations (shown as stars in Fig. 1), Banizoumbou (Niger, 13.54°N, 2.66°E) , Cinzana (Mali, 13.28°N, 5.93°W) , and M'Bour (Senegal; 14.39°N, 16.96°W), located on an east–west trajectory of the Sahara and Sahelian dust plumes (Sahelian Dust Transect) were deployed in the framework of African Monsoon Multidisciplinary Analysis (AMMA, Redelsperger et al., 2006; Marticorena et al., 2010) international project in 2006. The stations monitor surface particulate matter 10 (PM$_{10}$; with aerodynamic diameter ≤ 10 μm) concentrations, which are mainly dust concentrations, and local meteorological conditions over the Sahel (Marticorena et al., 2010). All the data are maintained by the Laboratoire Interuniversitaire des Systèmes Atmosphériques (Interuniversity Laboratory of Atmospheric Systems; LISA) in the framework of the International Network to study Deposition and Atmospheric composition in Africa (INDAAF; Service National d'Observation de l'Institut National des Sciences de l'Univers, France). Hourly observations of PM$_{10}$ concentrations   and

surface wind speed and precipitation from 2008 to 2020 are used to understand the day-night differences in dust aerosols and the potential impacts of meteorological conditions on the day-night differences.

### 2.2.5 Reanalysis Datasets

We also compare DOD from MERRA-2 (Gelaro et al., 2017; Randles et al., 2017) and EAC4 (Inness et al., 2019) global aerosol reanalysis datasets with IASI DOD. MERRA-2 is the first long-term (1980–present) reanalysis product in which aerosol and meteorological observations are jointly assimilated into global assimilation systems (Gelaro et al., 2017). It assimilates AOD from MODIS onboard Aqua and Terra, MISR, and Advanced Very High Resolution Radiometer (AVHRR) as well as observations from AERONET (Gelaro et al., 2017). EAC4 (Bozzo et al., 2017; Inness et al., 2019) is another aerosol reanalysis product we use in this study. It is produced using 4DVar data assimilation in ECMWF's Integrated Forecast System (IFS), and assimilates remote-sensed AOD from Envisat's AATSR and MODIS from Aqua and Terra (Bozzo et al., 2017). Hourly DOD from MERRA-2 and 3-hourly DOD from EAC4 from 2008 to 2020 are used.

Meteorological variables such as hourly surface winds, vertical velocity at 850 hPa, Convective Available Potential Energy (CAPE), and planetary boundary layer height (PBLH) from ECMWF Reanalysis v5 (ERA5; Hersbach et al., 2020) from 2008 to 2020 are used in this study. Similar variables from MERRA-2 are also used for a comparison. Here, we resample the meteorological data at each grid point based on IASI overpass time so at each grid point the meteorological variables are at the same time as IASI retrievals. For the full diurnal cycle, variables are shifted to local solar time (LST) based on the longitude of each AERONET site.

### 2.2.6 IMERG-GPM

Precipitation from the Integrated Multisatellite Retrievals for Global Precipitation Measurements (IMERG; Huffman et al., 2015) from 2008 to 2020 is used to examine the impacts of precipitation on the day-night differences in dust aerosols over the dust belt. IMERG builds upon the legacy of the Tropical Rainfall Measuring Mission (TRMM) by providing high quality estimates of global rainfall and snow for every 30 minutes at 10 km spatial resolution. The "Final Run" product of IMERG (version V06B), which is calibrated with Global Precipitation Climatology Centre (GPCC) reanalysis product, is used in this study. IMERG has been extensively validated against gauge, gridded, and satellite precipitation products over Africa (Dezfuli et al., 2017; Maranan et al., 2020; Ageet et al., 2022), the Middle East (Hosseini-Moghari and Tang, 2020; Arshad et al., 2021), and Asia (Huang et al., 2018; Kim et al., 2017; Lee et al., 2019). Though the performance of IMERG varies both spatially and temporally, it is shown by these studies to reasonably capture the observed precipitation over the dust belt. Some of the limitations of IMERG include large biases over mountainous areas (Huang et al., 2018), proneness to low-intensity false alarms and overestimation of rainfall amount in weak convective systems over the West African forest zone (Maranan et al., 2020).

### 2.3 Validation of IASI DOD against AERONET station observations

IASI daytime 10 $\mu$m DOD (LMD version) has been validated against AERONET solar CAOD by Capelle et al. (2014, 2018) at some selected AERONET sites over land and ocean for 2007–2016. In this work, we extend the previous daytime validation by including nighttime retrievals over the dust belt and to a longer period from 2007 to 2020. To compare IASI DOD with AERONET CAOD, we first sample AERONET solar and lunar CAOD within ±30 minutes of IASI overpass time and IASI DOD pixels within a radius of 30 km from the AERONET sites. In total, 22,462 and 944 AERONET-IASI

matchups for daytime and nighttime, respectively, are used in this study. Next, we convert IASI DOD at 10 $\mu$m in the infrared band (IR) to 500 nm in the visible band (VIS) to be consistent with AERONET CAOD at 500 nm. An accurate conversion requires detailed information of the refractive index, size distribution, and the effective radius of dust particles (Capelle et al., 2014), which are usually not available over a large domain. Previously, Peyridieu et al. (2013) and Capelle et al. (2014, 2018) compared IASI DOD with AERONET station data by scaling AEROENT AOD (550 nm) or CAOD (500 nm) to 10 $\mu$m using

empirically determined IR/VIS ratio at each AERONET site. Here, we follow a similar approach. At each AERONET station, the IR/VIS ratio is determined by regressing AERONET CAOD onto IASI DOD, with the slope of the regression being the IR/VIS ratio. However, the quality of such a linear fit depends on the sample size of IASI-AERONET collocations (Capelle et al., 2014). To prevent the ratios from being biased by the sample size, we exclude sites with number of IASI-AERONET collocations less than 100 for solar observations and 60 for lunar observations. Out of the 46 AERONET solar sites considered,

only 5 sites (CATUC_Bamenda, Zinder_Airport, Banizoumbou, LAMTO-STATION, and NAM_CO) were excluded whereas 4 out of the 11 lunar sites (Koforidua_ANUC, CATUC_Bamenda, Teide, and DEWA_ResearchCentre) were also excluded (see Tables 2 and 3). We found a mean IR/VIS ratio of ~0.62 ranging from 0.31 to 2.06 for solar measurements, and ~0.57 ranging from 0.26 to 1.23 for lunar observations. A constant IR/VIS ratio of 0.60 (approximated by taking the mean of 0.62 for solar and 0.57 for lunar) is used to scale all IASI DOD from IR to VIS equivalent DOD at 500 nm for both data validation and

the day-night difference analysis. Although the simple conversion method used here may lead to some uncertainties in the magnitude of the converted 500 nm IASI DOD, we found the calculated ratios to be largely within the range of empirically estimated IR/VIS ratios by Peyridieu et al. (2013) and Capelle et al. (2014, 2018) and largely consistent with the VIS/IR ratios used to convert IASI DOD (e.g., 1.54 by Yu et al. (2019) and 2.0 by Clarisse et al. (2019) for the conversion of IASI 10 $\mu$m DOD to 550 nm).

**Table 2. AERONET solar sites used in this study with their location and the short names labelled on figures. Also shown are the infrared to visible conversion ratios (IR/VIS) of each AERONET site for the solar measurements, the correlation coefficient (r) between IASI and AERONET CAOD at 500 nm, number of IASI-AERONET collocated data points (N), relative bias (%), and root mean square error (RMSE). All the correlation coefficients pass the 95% confidence level. The sites are divided into three broad regions of the dust belt: North Africa (NA), the Middle East (ME), and Asia (AS). Note that level 2 AERONET CAOD data are used for all solar sites except in Banizoumbou (Ban) and LAMTO-STATION (LAM) sites where level 1.5 data is used.**

| ID | Site | Short name | Long (°E) | Lat (°N) | IR/VIS | N | r | Bias (%) | RMSE | Region |
|---|---|---|---|---|---|---|---|---|---|---|
| 1 | Ben_Salem | Ben | 9.91 | 35.55 | 0.49 | 402 | 0.84 | 26.69 | 0.10 | |
| 2 | CATUC_Bamenda | CAT | 10.16 | 5.95 | 0.64 | 30 | 0.79 | 4.18 | 0.18 | |
| 3 | Dakar | Dak | -16.96 | 14.39 | 0.59 | 1062 | 0.79 | 23.26 | 0.20 | |
| 4 | IER_Cinzana | Cin | -5.93 | 13.28 | 0.49 | 129 | 0.82 | 41.69 | 0.19 | |
| 5 | Ilorin | Ilo | 4.67 | 8.48 | 0.40 | 557 | 0.82 | 25.81 | 0.24 | |
| 6 | Izana | Iza | -16.5 | 28.31 | 1.23 | 806 | 0.78 | -224.55 | 0.29 | |
| 7 | Koforidua_ANUC | Kof | -0.3 | 6.11 | 0.31 | 237 | 0.76 | 40.90 | 0.36 | North African sites (NA) |
| 8 | La_Laguna | Lag | -16.32 | 28.48 | 0.68 | 711 | 0.79 | -27.65 | 0.17 | |
| 9 | Lampedusa | Lam | 12.63 | 35.52 | 0.74 | 773 | 0.80 | -3.66 | 0.11 | |
| 10 | Medenine-IRA | Med | 10.64 | 33.5 | 0.45 | 311 | 0.75 | 35.61 | 0.07 | |
| 11 | Oujda | Ouj | -1.9 | 34.65 | 0.56 | 433 | 0.87 | 18.09 | 0.07 | |
| 12 | Santa_Cruz_Tenerife | SCT | -16.25 | 28.47 | 0.60 | 1333 | 0.78 | -0.20 | 0.13 | |
| 13 | Tamanrasset_INM | Tam | 5.53 | 22.79 | 0.45 | 228 | 0.52 | -243.93 | 0.57 | |
| 14 | Teide | Tei | -16.64 | 28.27 | 2.06 | 261 | 0.85 | -392.26 | 0.38 | |
| 15 | Zinder_Airport | Zin | 8.99 | 13.78 | 0.47 | 71 | 0.73 | -9.79 | 0.18 | |
| 16 | Banizoumbou | Ban | 2.67 | 13.55 | 0.39 | 67 | 0.69 | 35.66 | 0.23 | |
| 17 | LAMTO-STATION | LAM | -5.03 | 6.22 | 0.34 | 49 | 0.76 | 47.51 | 0.31 | |
| 1 | AgiaMarina_Xyliatou | Agi | 33.06 | 35.04 | 0.67 | 438 | 0.66 | -73.9 | 0.11 | |
| 2 | Antikythera_NOA | Ant | 23.31 | 35.86 | 0.81 | 225 | 0.80 | -15.05 | 0.09 | |
| 3 | Cairo_EMA_2 | Cai | 31.29 | 30.08 | 0.43 | 923 | 0.78 | 52.79 | 0.12 | |
| 4 | CUT-TEPAK | CUT | 33.04 | 34.67 | 0.64 | 926 | 0.76 | -4.3 | 0.08 | |
| 5 | DEWA_ResearchCentre | DEW | 55.37 | 24.77 | 0.43 | 169 | 0.72 | 43.02 | 0.13 | Middle East sites (ME) |
| 6 | Dhadnah | Dha | 56.32 | 25.51 | 0.42 | 146 | 0.64 | -6.31 | 0.16 | |
| 7 | Eilat | Eil | 34.92 | 29.5 | 0.44 | 942 | 0.31 | -130.95 | 0.33 | |
| 8 | Finokalia-FKL | Fin | 25.67 | 35.34 | 0.85 | 383 | 0.81 | -7.84 | 0.10 | |
| 9 | FORTH_CRETE | FOR | 25.28 | 35.33 | 0.63 | 562 | 0.80 | 11.58 | 0.07 | |
| 10 | Hada_El-Sham | Had | 39.73 | 21.8 | 0.58 | 162 | 0.83 | -65.2 | 0.17 | |
| 11 | KAUST_Campus | KAU | 39.1 | 22.3 | 0.55 | 1033 | 0.84 | 22.78 | 0.13 | |
| 12 | Kuwait_University | Kuw | 47.97 | 29.32 | 0.64 | 125 | 0.87 | -3.94 | 0.21 | |

| ID | Site | Short name | Long | Lat | IR/VIS | N | r | Bias | RMSE | Region |
|---|---|---|---|---|---|---|---|---|---|---|
| 13 | Masdar_Institute | Mas | 54.62 | 24.44 | 0.53 | 730 | 0.82 | 29.46 | 0.13 | |
| 14 | Mezaira | Mez | 53.75 | 23.10 | 0.42 | 1094 | 0.76 | 10.44 | 0.14 | |
| 15 | Migal | Mig | 35.58 | 33.24 | 0.46 | 340 | 0.64 | 14.51 | 0.11 | |
| 16 | Mussafa | Mus | 54.47 | 24.37 | 0.53 | 134 | 0.76 | 23.17 | 0.18 | |
| 17 | Nes_Ziona | Nes | 34.79 | 31.92 | 0.50 | 404 | 0.84 | 22.68 | 0.10 | |
| 18 | Nicosia | Nic | 33.38 | 35.14 | 0.55 | 294 | 0.69 | -15.61 | 0.08 | |
| 19 | Qena_SVU | Qen | 32.75 | 26.20 | 0.46 | 148 | 0.81 | 45.19 | 0.11 | |
| 20 | SEDE_BOKER | SED | 34.78 | 30.86 | 0.35 | 1642 | 0.72 | 58.02 | 0.08 | |
| 21 | Shagaya_Park | Sha | 47.06 | 29.21 | 0.47 | 423 | 0.73 | -11.26 | 0.13 | |
| 22 | Solar_Village | Sol | 46.4 | 24.91 | 0.51 | 671 | 0.87 | 43.66 | 0.16 | |
| 23 | Technion_Haifa_IL | Tec | 35.02 | 32.78 | 0.65 | 231 | 0.84 | -11.94 | 0.07 | |
| 24 | Weizmann_Institute | Wei | 34.81 | 31.91 | 0.61 | 515 | 0.81 | 31.39 | 0.08 | |
| 1 | Jaipur | Jai | 75.81 | 26.91 | 0.64 | 771 | 0.88 | 11.04 | 0.10 | Asian sites (AS) |
| 2 | Karachi | Kar | 67.14 | 24.95 | 0.68 | 810 | 0.89 | 32 | 0.14 | |
| 3 | MCO-Hanimaadhoo | MCO | 73.18 | 6.78 | 0.41 | 619 | 0.56 | 18.81 | 0.07 | |
| 4 | Nainital | Nai | 79.46 | 29.36 | 1.36 | 127 | 0.92 | -158.02 | 0.27 | |
| 5 | NAM_CO | NAM | 90.96 | 30.77 | 0.39 | 15 | 0.18 | -422.13 | 0.22 | |

**Table 3. Same as Table 2 but for AERONET lunar measurements. Sites with asterisk denotes insignificant correlation coefficient at the 95% confidence level.**

| ID | Site | Short name | Long (°E) | Lat (°N) | IR/VIS | N | r | Bias (%) | RMSE | Region |
|---|---|---|---|---|---|---|---|---|---|---|
| 1 | Ilorin | Ilo | 4.67 | 8.48 | 0.26 | 66 | 0.44 | -0.59 | 0.35 | |
| 2 | Koforidua_ANUC | Kof | -0.30 | 6.11 | 0.29 | 53 | 0.58 | 29.64 | 0.39 | |
| 3 | CATUC_Bamenda* | CAT | 10.16 | 5.95 | 0.06 | 8 | 0.14 | 23.71 | 0.52 | North Africa |
| 4 | Teide | Tei | -16.64 | 28.27 | 2.47 | 57 | 0.71 | -561.83 | 0.17 | |
| 5 | Dakar | Dak | -16.96 | 14.39 | 0.67 | 88 | 0.73 | -7.06 | 0.21 | |
| 6 | Izana | Iza | -16.5 | 28.31 | 1.23 | 80 | 0.69 | -221.28 | 0.11 | |
| 7 | Santa_Cruz_Tenerife | SCT | -16.25 | 28.47 | 0.28 | 71 | 0.82 | 53.22 | 0.17 | |
| 1 | Shagaya_Park | Sha | 47.06 | 29.21 | 0.39 | 144 | 0.68 | 20.64 | 0.19 | Middle East |
| 2 | Mezaira | Mez | 53.75 | 23.10 | 0.67 | 206 | 0.56 | -39.91 | 0.16 | |
| 3 | Migal | Mig | 35.58 | 33.24 | 0.49 | 114 | 0.47 | -9.86 | 0.11 | |
| 4 | DEWA_ResearchCentre | DEW | 55.37 | 24.77 | 0.37 | 57 | 0.27 | -6.91 | 0.20 | |

## 3. Results

### 3.1 Evaluation of daytime and nighttime IASI DOD against AERONET CAOD

We evaluate IASI daytime and nighttime DOD against AERONET ground observations before using the product to understand the day-night differences in dust aerosols over the dust belt. Such evaluations can be achieved by Taylor diagrams (Taylor, 2001). A Taylor diagram compares datasets in terms of three statistics i.e., the Pearson correlation coefficient between the two datasets, the standard deviations, and the centered root mean square error (RMSE). Figure 2 shows normalized Taylor diagrams that compare IASI DOD (scaled to 500 nm using the average IR/VIS ratio of 0.60) to AERONET CAOD (500 nm) for daytime (Fig. 2a) and nighttime (Fig. 2b) observation. The standard deviations and centered RMSEs of IASI DOD have been normalized by the standard deviation of AERONET CAOD (shown as REF). The results show IASI DOD is highly correlated with AERONET station observations with statistically significant (95% confidence level) correlation coefficients ranging between 0.18–0.92 for solar sites and 0.14–0.82 for lunar sites. The highest average correlation coefficient for solar data is observed in the Saharan and Sahelian dust belt with correlation coefficient of 0.77 ranging from as low as 0.52 in Tamanrasset (Tam) to as high as 0.87 in Oudjda (Ouj), followed by the Middle East sites with an average correlation coefficient of 0.75 ranging from 0.31 in Eilat (Eil) to 0.87 in Solar Village (Sol) and Kuwait University (Kuw). The performance of IASI over the Asian sites is highly variable with the lowest correlation coefficient of 0.18 in NAM_CO (NAM) site to as high as 0.92 in Jaipur (Jai). The NAM site has the lowest sample size of IASI-AERONET collocations (N=15), and this may partly account for such a low correlation coefficient. These results are largely consistent with similar evaluations in past studies (Peyridieu et al., 2013; Capelle et al., 2014, 2018). However, we also notice underestimation of daytime DOD at some few sites, such as Eilat (Eil) with a weaker correlation coefficient of 0.36, higher RMSE of 0.33, and a large bias of more than -100% (see Table 2 and Fig. 2a). Similar large biases are also observed around other coastal sites over North Africa (e.g., Iza, Lag, and Tei) possibly due to the mixing of sea salt with dust aerosols and the complicated land surface conditions in the area leading to difficulties in DOD retrieval (Capelle et al., 2014, 2018). Similarly, nighttime DOD is also underestimated at Tei and Iza by more than -200%

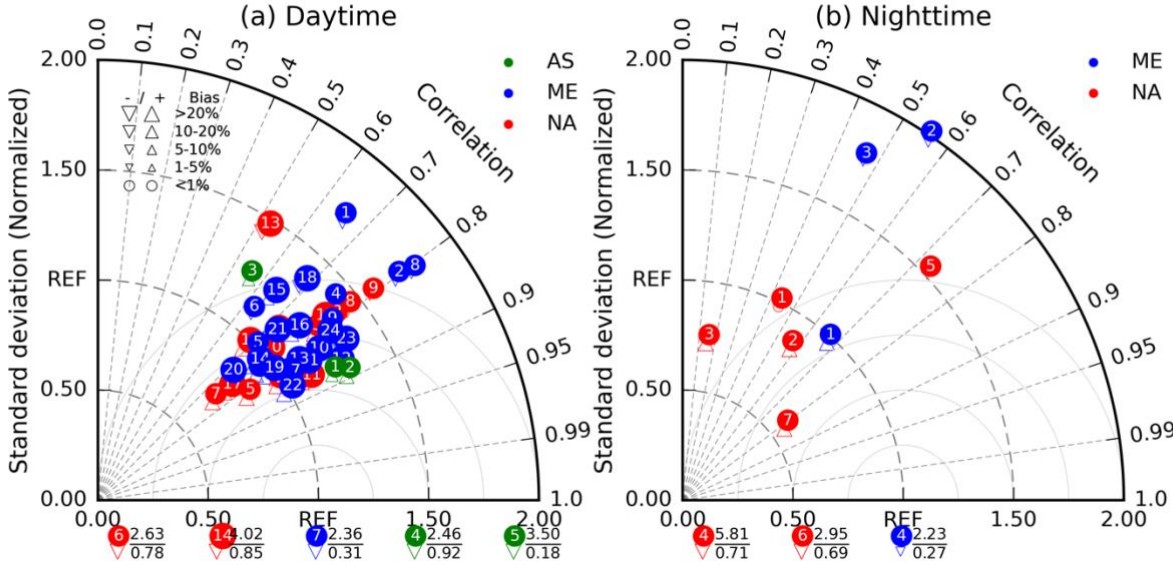

**Figure 2: Normalized Taylor diagrams for IASI DOD and AERONET CAOD at 500 nm during (a) daytime (9:30 a.m. local solar ECT) and (b) nighttime (9:30 p.m. local solar ECT). AERONET CAOD is sampled between ±30 minutes of IASI overpass time, and IASI DOD is sampled within a radius of 30 km from each AERONET site. The grey-dashed semi circles show the normalized standard deviations, grey-solid semi circles denote the normalized centered RMSE, and the dashed radial lines represent the Pearson correlation coefficients. Sites are identified by their ID in Tables 2 (daytime) and 3 (nighttime), denoted by the number in the colored circles. Red, blue, and green denotes sites in North Africa (NA), the Middle East (ME), and Asia (AS), respectively. Relative biases are denoted by triangle, with upward (downeard) triangles indicating a positive (negative) bias. Sites with normalized standard deviation greater than 2.0 are shown at the bottom of the Taylor diagram. Numbers above the black line are the normalized standard deviations and below are correlation coefficients.**

The correlations over lunar sites are generally lower than solar sites (Fig. 2b). While over sites like Teide (Tei), Dakar (Dak), and Santa Cruz Tenerife (SCT) where correlations between IASI DOD and AERNOET CAOD are higher than 0.7, correlations over other sites are around 0.14~0.69, with the lowest correlation of 0.14 at CATUC_Bamenda (CAT) site. Note that the smaller correlation coefficient at CAT is insignificant and may be due to the complex topography of the area that makes IASI retrieval difficult resulting in smaller IASI-AERONET collocated sample size (N=8). Some sites over the Middle East (e.g., DEWA_Research_Centre (DEW) and Migal (Mig)) are also characterized by slightly lower nighttime correlation coefficients. The discrepancies between IASI DOD and AERONET CAOD records at sites over complex topographic regions (e.g., Iza with an altitude of ~2.4 km) are also observed by Capelle et al. (2008) who attributed such lower correlations partially to the heterogeneity of land surface or rapid varying near-surface dust plume that may reduce the sensitivity of infrared sounders. Reasons for the lower correlation in lunar data could range from smaller sample size of lunar data to the quality of data used in the evaluation, which are cloud screened but not quality assured. In general, IASI DOD at sites around dust source regions is better correlated with AERONET CAOD than sites around regions where dust is transported from source regions (e.g., the

southern Sahel), and worsened in areas characterized by complex terrains and pollutants from either biomass burning, industrial emissions or coastal sediments (e.g., Eilat).

In addition to these Taylor diagrams, we further examined the relationship between IASI DOD and AERONET CAOD by combining all data points for daytime and nighttime measurements as shown in Fig. 3a–b. Consistent with the Taylor diagrams, the density scatter plots reveal a good performance of IASI DOD with overall correlation coefficient of 0.7 for solar

observations (Fig. 3a) and 0.57 for lunar measurements (Fig. 3b). These values are quite close to the average correlations over all solar sites in Table 2 (0.75) and all lunar sites in Table 3 (0.55). We also notice there are some overestimations of IASI DOD for small CAOD values (<0.5) in both daytime and nighttime records (mainly over coastal sites such as, Mez, Eil, and Dak in daytime and Mig, Mez, and Bho at nighttime), which warrant future investigation. Overall, Figs. 2-3 show LMD IASI well captures the spatiotemporal distribution of dust aerosols over the dust belt in both daytime and nighttime can therefore be used

to understand the day-night variations in dust aerosols.

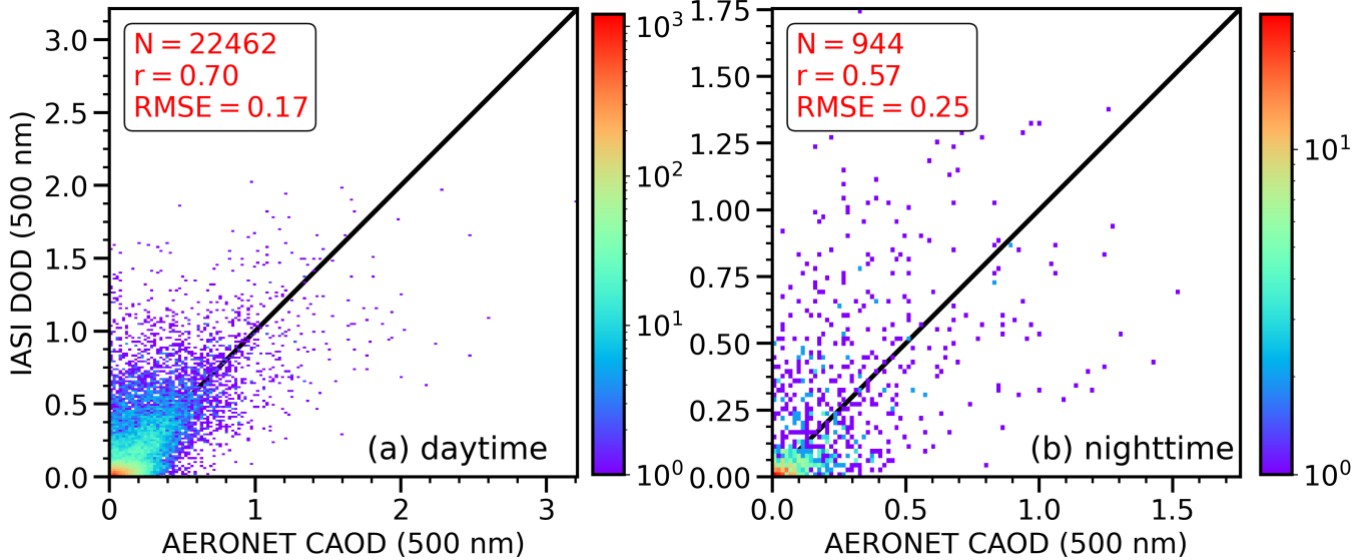

Figure 3: **A bivariate histogram (log scale) of IASI DOD versus AERONET CAOD over (a) all the 46 AERONET solar sites and (b) 11 lunar sites across the dust belt (see locations in Tables 2 and 3, and Fig. S1).** *r* **is the correlation coefficient between IASI DOD (scaled to 500 nm) and AERONET 500 nm CAOD,** RMSE **is the root mean square error, and** *N* **is the sample size of IASI-AERONET collocations.**

**3.2 Characteristics of daytime and nighttime dust activities over the dust belt**

In this section, we examine the characteristics and differences between daytime (9:30 a.m. local solar ECT) and nighttime (9:30 p.m. local solar ECT) DOD and dust layer height from LMD IASI, along with CAOD from nine selected AERONET stations and surface $PM_{10}$ concentrations from three LISA sites. Here, a mean uniform scaling factor of 0.60 is

used to convert both daytime and nighttime 10 μm DOD to 500 nm. Using individual ratios will slightly improve the consistency between IASI DOD and AERONET CAOD (not shown) but may lead to some biases in the day-night differences. Figure 4 shows the annual and seasonal mean climatology (2008 to 2020) of IASI 500 nm DOD, AERONET 500 nm CAOD, and surface $PM_{10}$ concentration for daytime, nighttime, and day-night difference. Both daytime and nighttime DOD show similar seasonal cycles. In winter (DJF), the dustiest regions occur in the southern parts of the Sahel to the Guinea Coast (Fig. 4b, g). By spring (MAM), the maximum DOD begins to transition northward to the central to northern parts of the Sahara (Fig. 4c, h) and maximizes around summer (JJA) in the central to the northwestern Sahara (Fig. 4d, i). similarly, a pronounced DOD maximum is observed in the central parts of the Arabian Peninsula, northwestern parts of the Indian subcontinent, around the Iraqi and Irani deserts, and the Taklamakan Desert in northwestern China in JJA. DOD reduces in fall (SON), with a magnitude comparable to that in DJF over the Middle East and Asia, but slightly stronger over the Sahara Desert yet weaker over the Sahel (Fig. 4e, j). Such seasonal migration of dust maxima is largely driven by the meridional migration of the Intertropical Convergence Zone (ITCZ) and generally consistent with previous studies about dust aerosols in this region via satellite retrievals (e.g., Ginoux et al., 2012; Pu and Ginoux, 2018; Yu et al., 2019; Chédin et al., 2020; Vandenbussche et al., 2020; Yu et al., 2021; Li et al., 2021).

Figure 4 also demonstrates statistically significant (95% confidence level) differences between daytime and nighttime DOD. The day-night differences in DOD, i.e., daytime minus nighttime, are positive over the major dust source regions (i.e., most parts of the Sahara, the central Arabian Peninsula, parts of South Asia around East Iran, southwest Afghanistan, and central Pakistan, and the Taklamakan Desert) yet negative over regions near dust sources (i.e., the southern Sahel to the Guinea Coast, the southeastern coast of the Arabian Peninsula, and central to southern India). It is also important to note that there is a seasonal variability in the magnitude of the day-night differences in DOD, with the largest magnitude of the day-night difference in DOD in DJF and MAM (Fig. 4l, m) and a weaker magnitude in JJA and SON (Fig. 4n, o). The spatial pattern of the day-night differences in DOD in JJA is generally consistent with the day-night difference in dust emissions over North African dust sources shown by Chédin et al. (2020; e.g., their Fig. 4) and Todd and Cavazos-Guerra (2016; e.g., their Fig. 8).

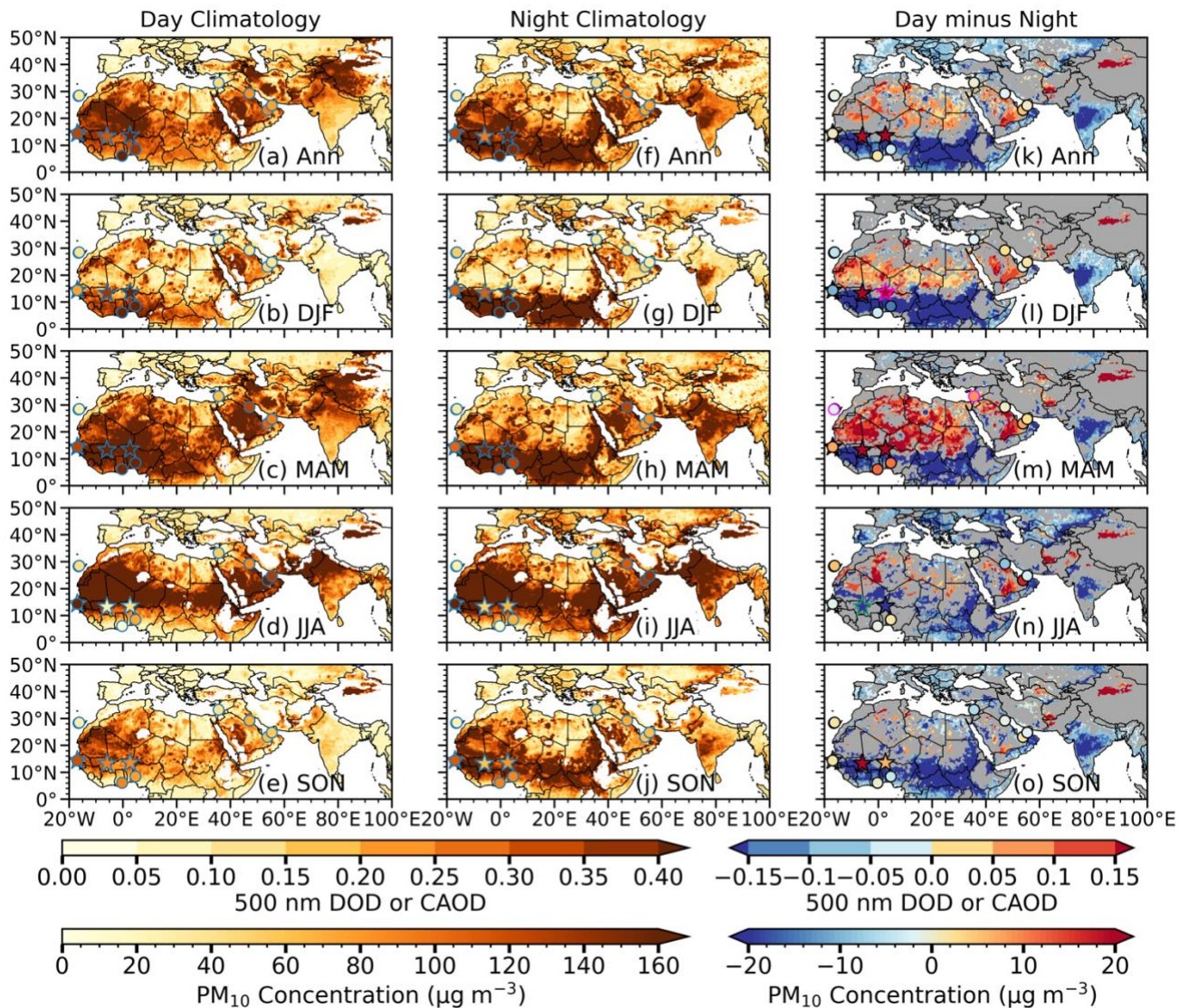

**Figure 4: Annual (Ann) and seasonal means of LMD IASI DOD (scaled from 10 μm to 500nm using an IR/VIS ratio of 0.60) from 2008 to 2020 at (a)–(e) daytime (9:30 a.m. ECT), (f)–(j) nighttime (9:30 p.m. ECT), and (k–o) day-night differences, along with LISA PM$_{10}$ concentrations averaged over 2008–2020 and AERONET CAOD overlayed as stars and dots, respectively. The white color denotes oceanic grid cells and missing values over land. In (k–o), areas where day-night differences in DOD do not pass the 95% confidence level (t-test) are masked in grey. The magenta and green colors around the edges of LISA and AERONET sites in (k–o) show sites where the day-night differences in CAOD or PM$_{10}$ concentrations pass the 95% and 90% confidence levels repectively.**

Surface observations of dust properties are also examined to compare with results from IASI products. AERONET
CAOD overlayed as circles on the IASI DOD show the two datasets generally agree on the seasonal daytime and nighttime

distributions of dust aerosols (Fig. 4a–j), but the day-night differences in most seasons are insignificant, partially due to smaller sample size and lower quality (level 1.5) of lunar data. Among all the sites analyzed, only Iza and Mig sites show significantly positive day-night differences in CAOD in MAM, with Mig largely consistent with IASI DOD. The inconsistency between IASI DOD and AERONET CAOD in some AERONET sites may also be partially due to the uncertainties resulting from using
CAOD to approximate DOD and uncertainties in IASI retrieval (see discussion in section 3.7) as well.

Surface observations of $PM_{10}$ concentrations at both daytime and nighttime from LISA are also overlayed as stars on the IASI DOD in Fig. 4. For consistent comparison with IASI DOD, LISA hourly data is averaged over timesteps approximately within ±30 minutes of IASI pixels that fall within 30 km radius from each LISA site. Although DOD and surface $PM_{10}$ concentrations reveal different aspect of dust activities, i.e., IASI DOD shows vertically integrated column extinction due to
coarse dust, while $PM_{10}$ concentrations reveal near-surface concentrations of large particles including both dust and sea salt (usually dominated by dust in dust source regions), we found that these results share similarities in terms of the day-night variations. For instance, the day-night difference in $PM_{10}$ over Cin in JJA and Ban in DJF (Fig. 4n, l; significant at the 90% and 95% confidence levels respectively) are quite consistent with IASI DOD.

IASI also retrieves dust layer height, a variable that can be useful in characterizing the day-night difference in the
distribution of dust aerosols. Figure 5 shows the annual and seasonal mean climatology of daytime, nighttime, and day-night differences in dust layer height. The dust layer height reaches about 2.4–3.6 km in dust source regions over the Sahara Desert and the Sahel, the central Arabian Peninsula, and the deserts in Central and East Asia in the annual mean (Fig. 5a, f), and are generally higher in DJF and MAM seasons (Fig. 5b, c, g, h) and lower in JJA and SON (Fig. 5e, j). The lower summertime dust layer height is somewhat in contrast to previous studies using CALIOP (e.g., Yu et al., 2010; Clarisse et al., 2019; See
Fig. S3 and more discussion below). Negative day-night differences in dust layer height, i.e., lower dust layer height at daytime than nighttime, are observed mainly in dust source regions (e.g., large parts of the Sahara Desert, Arabian Peninsula, and the Taklamakan Desert), while positive differences are found over the dust downwind regions (e.g., the southern Sahel to the Guinea Coast and large areas in the Indian subcontinent (Fig. 5k–o). The magnitude of the day-night differences in dust layer height shows relatively small seasonal variations. Overall, the spatial pattern of the day-night differences in dust layer height
(Fig. 5k–o) is largely opposite to that of DOD (Fig. 4k–o), which is generally consistent with the dust emission index defined by Chédin et al. (2020) that shows higher DOD and lower dust layer height in dust source regions.

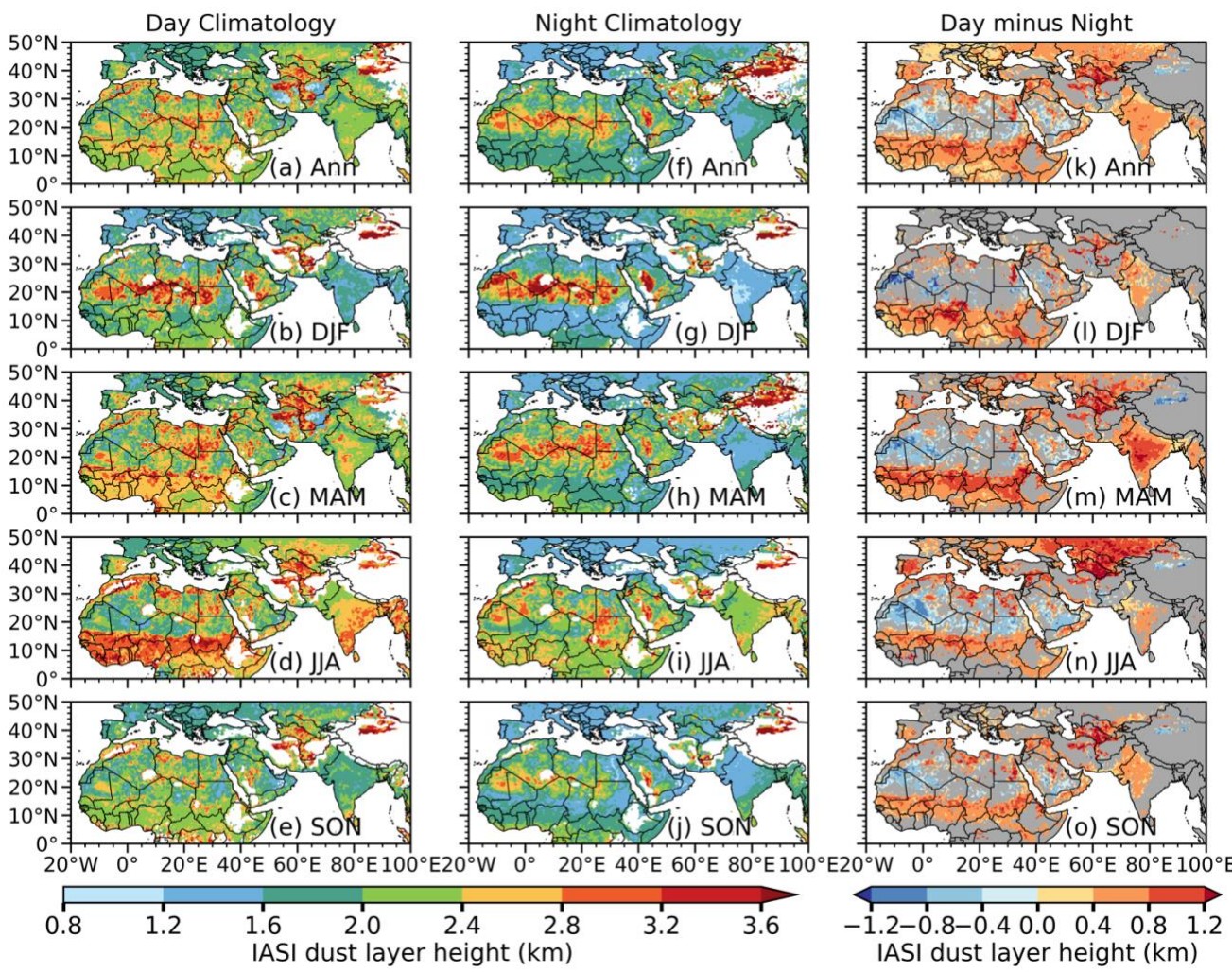

**Figure 5: Same as Fig. 4., but for dust layer height (km) from IASI. Dust layer height is defined as the height where half of the vertically integrated DOD is above, and the other half is below.**

The seasonal cycle of daytime (1:30 p.m. local solar ECT) and night (1:30 a.m. local solar ECT) DOD and dust layer
height from CALIOP is also investigated to compare with IASI data (Figs. S2 and S3, respectively). The seasonal cycle of
CALIOP DOD is very similar to IASI, consistent with the findings of Yu et al. (2019), although IASI shows a larger area of
high DOD over the Guinea coast at nighttime in DJF (Figs. 4g, S2g). The day-night differences in DOD from CALIOP are
insignificant for most parts of the dust belt across all the seasons except for a narrow region over the northern Sahara and parts
of central Asia and western China in MAM, JJA and SON and are in general opposite to IASI. Such an inconsistency in the
day-night differences in DOD between IASI and CALIOP may partially be attributed to the low signal-to-noise ratio of
CALIOP daytime data and differences in the overpass times of the two instruments (~4 hours apart). Moreover, because of the

narrow swath width of CALIOP with a sampling rate of twice per month (repeat cycle of 16 days), the afternoon and nighttime observations are not at the same day, which also influence the day-night differences in CALIOP DOD.

In contrast to IASI, dust layer height in CALIOP shows maximum altitude in JJA over major dust source regions and
minimum in DJF (Fig. S3). Previous studies show IASI dust layer height is systematically biased low by about –0.4 to –0.8 km in comparison with CALIOP (Peyridieu et al., 2013; Capelle et al., 2014; Kylling et al., 2018), however, here we found that the maximum altitudes are comparable between the two datasets over North Africa. The afternoon (1:30 p.m. local solar ECT) and midnight (1:30 a.m. local solar ECT) dust layer height from CALIOP is also not significantly different from each other in most parts of the dust belt, except around the southern Sahel to the Guinea Coast, southern Arabian Peninsula, and parts of the
western Taklamakan Desert in DJF, western China in MAM, northern and eastern Sahara and part of central Asia in JJA, and western China in SON (Fig. S4l–o), while sharing some similarity with IASI over the western Taklamakan Desert, central Asia (JJA), and coastal northwestern Africa (JJA). The differences between IASI and CALIOP dust layer height could be attributed to several factors (Peyridieu et al., 2013; Chédin et al., 2020), such as different definitions of dust layer height, e.g., arithmetic mean dust layer height in CALIOP versus cumulative extinction height in IASI, and different overpass times of the two
instruments (CALIOP lags IASI by about 4 hours). Kylling et al. (2018) found that the bias of dust layer height in IASI (LMD version) would be lower if CALIOP dust layer height was defined by cumulative extinction height instead of arithmetic mean and was shifted to the observation time of IASI. Their results (their Table 3) show a difference of –0.053±1.339 km between LMD IASI and CALIOP for the cumulative extinction and –0.607±1.187 km for the arithmetic mean.

## 3.3 Seasonal cycle of day-night variations in dust aerosols from IASI, LISA, and AERONET

We compare the seasonal cycle of daytime and nighttime IASI DOD with LISA and AERONET ground-based observations to better understand how day-night differences in dust aerosols propagate in seasons. Figure 6 shows monthly mean surface $PM_{10}$ concentrations from three LISA sites (Ban, Cin and Mbo; see locations in Fig. 1) and monthly mean DOD from IASI averaged over a 30 km radius around LISA sites. We average hourly $PM_{10}$ concentrations around ±30 minutes of IASI overpass time for a consistent comparison with IASI. Note that the seasonal cycle of LISA records is different from DOD,
with a minimum in JJA associated with monsoon rainfall and a peak in DJF and MAM due to transported dust from the central Sahara (Marticorena et al., 2010). The three sites are along 13–14° N but aligned in an east-west trajectory of the Sahelian Dust Transect. Such observations reveal a clear spatial variability of dust with higher dust concentration over Banizoumbou (Ban) which is close to the Saharan dust sources but decreases westward in Cinzana (Cin) and M'Bour (Mbo), similar to the findings of previous studies (Marticorena et al., 2010; Kaly et al., 2015). Daytime $PM_{10}$ concentration is significantly higher than
nighttime in DJF and early MAM at Ban and Cin, while nighttime dust concentration is higher than daytime from late MAM to early SON (or late JJA) at Ban and Cin (Fig. 6a-b). Mbo shows similar seasonal cycle as Ban and Cin but the day-night difference is largely insignificant (Fig. 6c). Like LISA $PM_{10}$, daytime IASI DOD is higher than nighttime in most of DJF and MAM months but lower in JJA at Ban and Cin, consistent with the results shown in Fig. 4. Note that different from $PM_{10}$ concentrations, nighttime IASI DOD at Mbo is higher than daytime throughout the entire year. This disparity could partially

be due to the fact that Mbo site is located along the transport path of the boreal JJA dust plumes, but further from the major

dust sources of North Africa, thus dust aerosols are likely mixed to higher altitudes which may be sampled differently between

the near nadir viewing IASI instrument and the surface measurements, and differently between day and night.

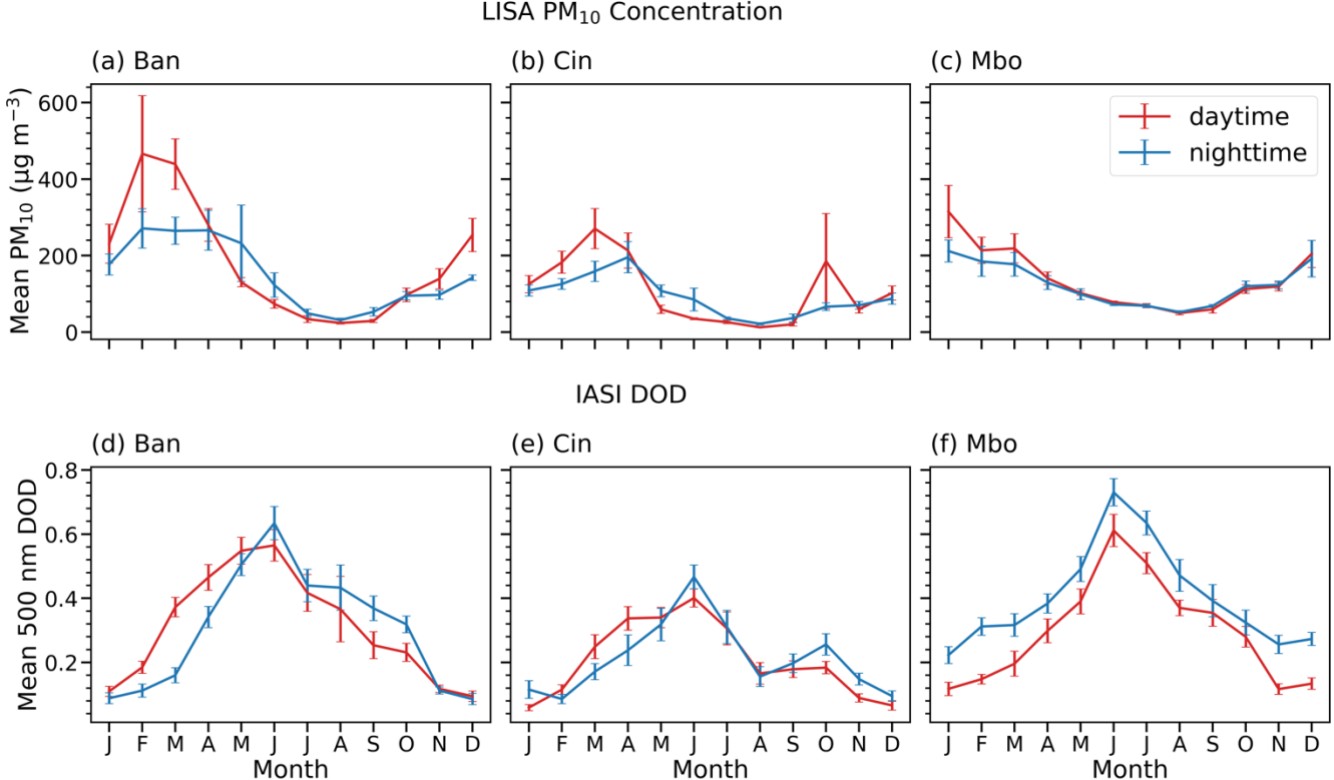

**Figure 6: Seasonal cycle of (a)–(c) $PM_{10}$ concentrations (µg m$^{-3}$) from LISA sites and (d)–(f) DOD from IASI at the same locations averaged from 2008 to 2020 except for LISA $PM_{10}$ in Mbo where the average is from 2008-2019. The error bars show standard errors. PM$_{10}$ concentrations and IASI DOD are collocated by averaging hourly LISA data around ±30 minutes of IASI overpass time and averaging IASI pixels that fall within a radius of 30 km from each LISA sites.**

      Similar analysis is carried out over nine AERONET sites (blue dots in Fig. 1) for IASI DOD and AERONET CAOD

as shown in Fig. 7. AERONET CAOD is collocated with IASI temporally and spatially for consistent comparison between the

two datasets. CAOD and DOD show very similar seasonal cycles, with maxima in late MAM to JJA for stations in the Sahara

(Dak) and off the west coast of North Africa (Iza, and SCT), and the Middle East (Mig, Sha, Mez, and DEW), whereas the

Guinea coast stations (Ilo and Kof) showed maximum DOD or CAOD in the late DJF to MAM. The largest biases between

IASI and AERONET occur in Iza during JJA where both IASI daytime and nighttime DOD overestimate AERONET solar and

lunar measurements. It is worth noting that Iza site is located at higher altitude (~2.4 km) and may contain some uncertainties.

In terms of the spatial variability of dust, both IASI and AERONET showed consistency over all sites with maximum DOD or

CAOD in JJA over Dak, Iza, SCT, Mig, Mez, and DEW, maximum DOD or CAOD in DJF over Ilo and Kof, and maximum

DOD or CAOD in late MAM over Sha. In terms of the day-night differences, AERONET is consistent with IASI in Dak site during JJA, with higher CAOD at nighttime than daytime. Over the Guinea Coast (Ilo and Kof sites), nighttime CAOD or DOD is higher than daytime for most months from late JJA to DJF, which is consistent with IASI. While seasonal variations in day-night differences in CAOD are largely similar to IASI DOD at Dak, Ilo, Kof in JJA and SON, discrepancies are found at SCT in JJA, Mez in JJA and SON, Mig in MAM, Sha from MAM to SON, and DEW in MAM probably in association with the relatively smaller sample size and relatively lower quality (level 1.5) of AERONET lunar data and impacts of sea salt on CAOD at the coastal stations.

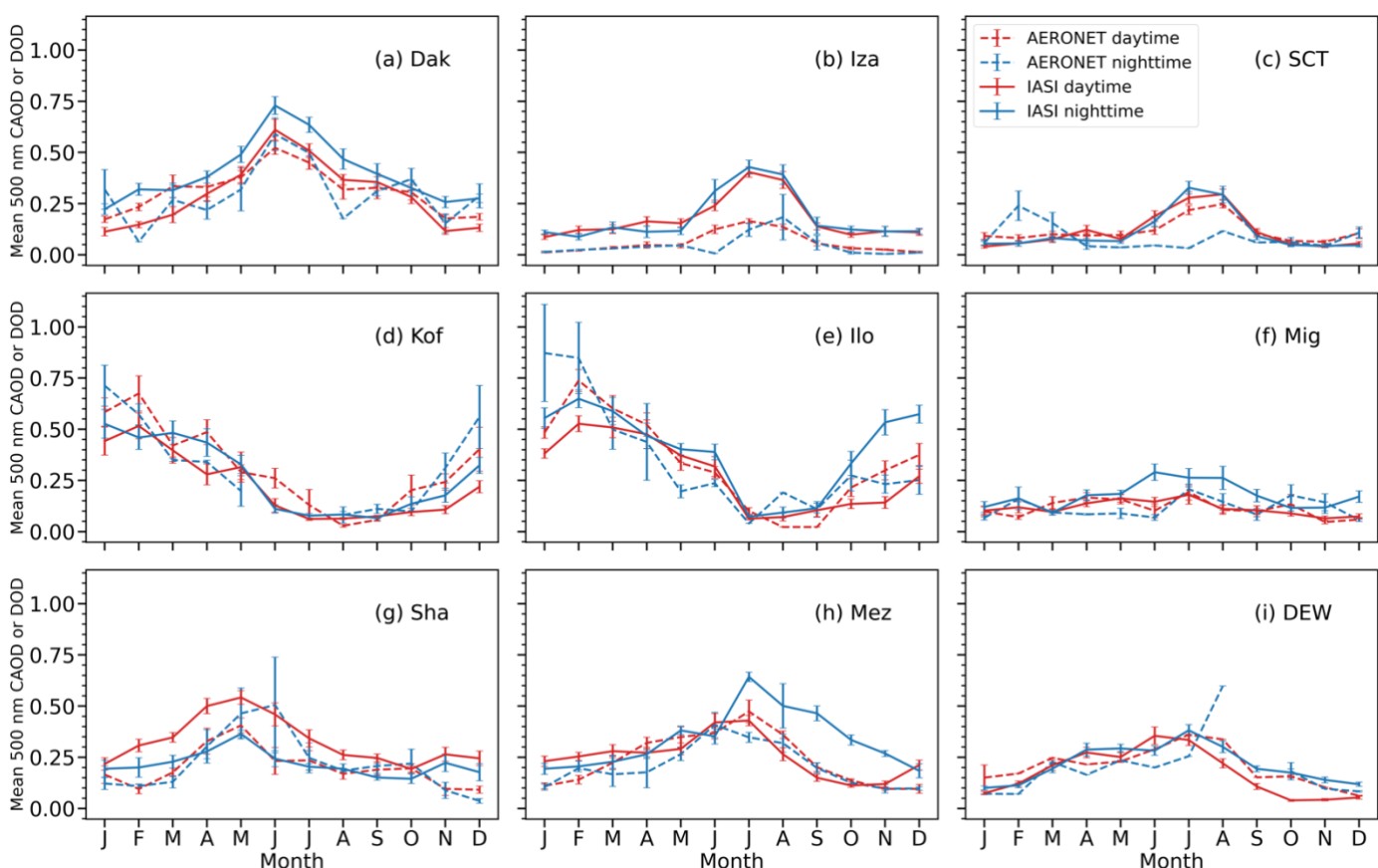

**Figure 7: Seasonal cycle of AERONET CAOD (dashed line) and LMD IASI DOD (solid line) at 500 nm for daytime (red) and nighttime (blue). The seasonal cycle of IASI DOD and AERONET CAOD were collocated and computed for a temporal range between 2008 and 2020. However, the temporal range of AERONET varies from site to site but within 2008 to 2020. AERONET solar and lunar observations are sampled ±30 minutes of IASI overpass time whereas IASI DOD is averaged over a radius of 30 km from each AERONET site.**

### 3.4 Diurnal variations in dust aerosols

IASI DOD and dust layer height are available only two times daily. To have a clear picture of the full diurnal cycle of dust aerosols, data with higher temporal resolution are required. Here we use station data, i.e., typically 5 to 15 minutes AERONET CAOD and hourly LISA $PM_{10}$ concentrations, to further explore diurnal variations in dust over the dust belt. Figure 8 represent the diurnal cycle of surface $PM_{10}$ concentration in the Sahel (first row) and CAOD at AERONET sites (last three rows) for each season, with cyan vertical lines highlighting the overpass time of IASI (9:30 a.m. and 9:30 p.m. ECT). The results indicate that surface $PM_{10}$ concentrations at the three LISA sites in the Sahelian dust belt peak around 9 a.m.–11 a.m. LST in the late morning in all seasons, except JJA and SON, when $PM_{10}$ concentrations are low due to precipitation scavenging. Both Cin and Mbo sites have evening peaks around 7–8 p.m. local solar time (LST) (except JJA at Mbo; Fig. 8b, c), but not very evident at Ban site, which shows a peak around mid-night in MAM (Fig. 8a). The passing time of IASI is largely consistent with the timing of maxima in surface $PM_{10}$ concentrations.

At AERONET sites, daytime records (6 a.m. to 5 p.m. for Dak, Ilo, and Kof sites; 5 a.m. to 6 p.m. for Iza, SCT, Mez, Mig, Sha, and DEW; light yellow shading in Fig. 8) are observed by sun photometer, and nighttime data (6 p.m. to 5 a.m. for Dak, Ilo, and Kof; 5 p.m. to 6 a.m. for Iza, SCT, Mez, Sha, DEW; 5 p.m. to 5 a.m. for Mig; grey shading in Fig. 8) are from lunar photometer, thus the discontinuity between daytime and nighttime records is likely due to the different instruments (Fig. 8d–l). Furthermore, level 1.5 lunar data also have higher uncertainty compared to level 2.0 solar data. AERONET CAOD also peaks in the morning around 7–9 a.m. LST for sites in the Guinea Coast (Ilo and Kof) in DJF, the Sahel (Dak) and the North Atlantic sites (Iza and SCT), and the Middle East (Mez) in JJA which is consistent with previous work in this region (Schepanski et al., 2009; Marticorena et al., 2010; Kaly et al., 2015; Yu et al., 2019; Yu et al., 2021). A secondary peak of CAOD occurs in the afternoon around 3 p.m. LST (e.g., at Dak, Kof, Ilo, and Mez sites; Fig. 8d, g, h, k), around 8 p.m. and 4 a.m. in Iza and SCT sites. The nighttime peak of CAOD varies in different regions. In North Africa, CAOD maximizes around 3 a.m. in Dak, 8 p.m. and 4 a.m. in Iza, 10 p.m. in SCT, 4 a.m. in Ilo, and midnight in Kof, while in the Arabian Peninsula CAOD peaks around 5 a.m. in Mig, 8 p.m. in Sha, and around 4 a.m. in DEW but without a clear peak in Mez site (slightly higher around 8 p.m. and 1 a.m. in JJA; Fig. 8d–l). Overall, the available AERONET data in the dust belt show that IASI 9:30 a.m. local solar ECT data largely captures the early morning peak of CAOD, while the 9:30 p.m. local solar ECT data partially captures the high CAOD either after its early evening peak or before its nighttime maxima. As revealed by the comparison with LISA and AERONET station data, the day-night variations in IASI DOD could be quite similar to ground observed CAOD and surface dust concentrations at some sites but not others. Although IASI data contains only two-time steps, its high spatial resolution and global coverage provide useful information that complements sparsely located ground observations to help understand the diurnal cycle of dust.

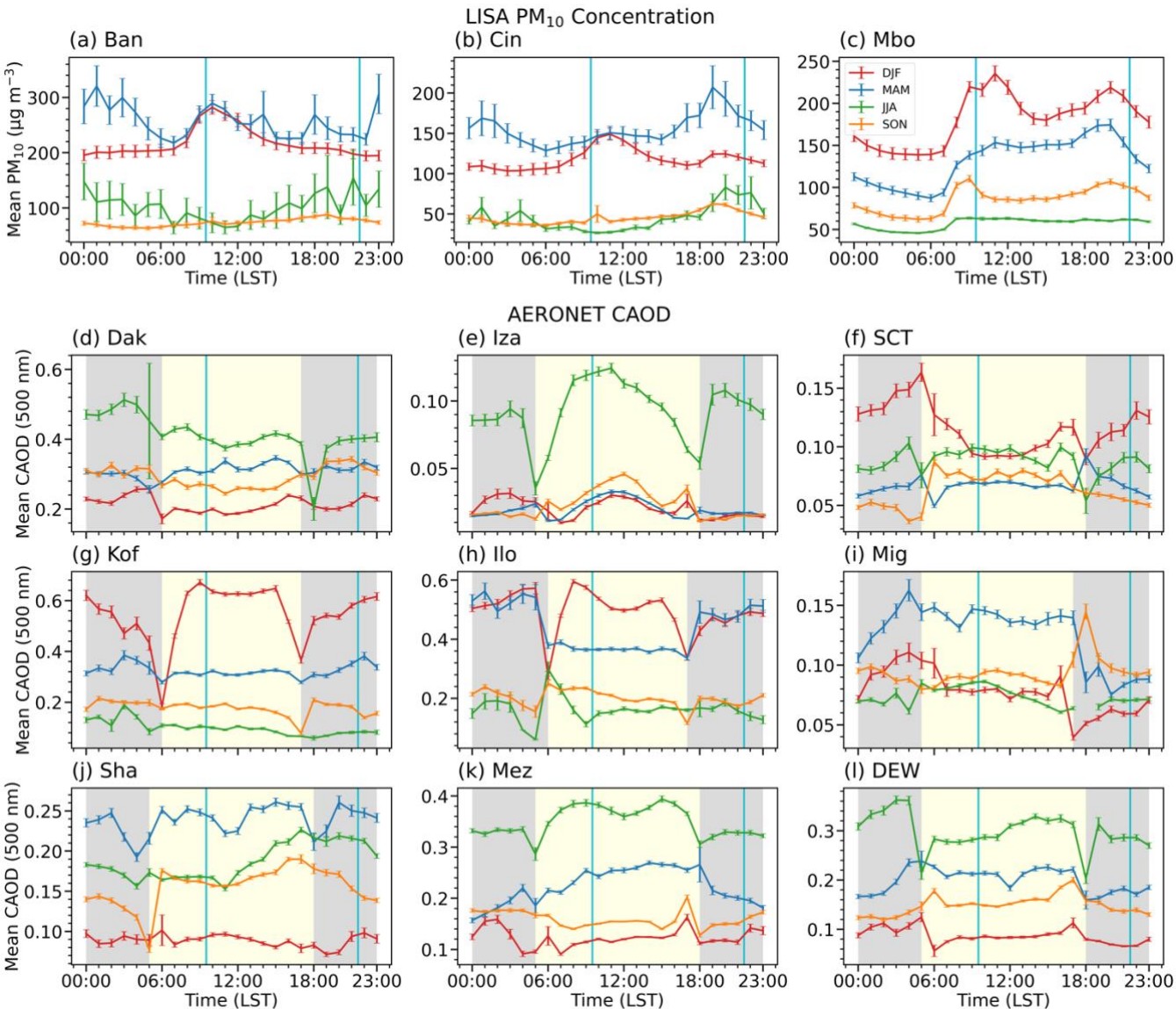

**Figure 8: Diurnal cycle of (a)–(c) LISA $PM_{10}$ concentrations (µg m$^{-3}$, a–c) in the Sahel and (d–i) AERONET CAOD over sites across the dust belt. The diurnal cycle of LISA $PM_{10}$ concentrations were averaged between 2008 and 2020 for Ban and Cin sites, and between 2008 and 2019 for Mbo site. The temporal ranges for AERONET data vary depending on the number of records available for both solar and lunar datasets. The cyan lines mark 9:30 a.m. and 9:30 p.m. local solar ECT. The grey (light yellow) background shading shows the temporal range of lunar (solar) observations. Error bars show standard errors.**

## 3.5 Daytime and nighttime DOD from reanalysis products

With global coverage and high temporal resolution, aerosol products from reanalysis would be great tools to study diurnal cycle of dust if they largely capture the observed day-night dust variations shown in satellite retrievals. Here we examine

daytime and nighttime DOD from MERRA-2 and EAC4 to examine whether they capture the day-night differences in DOD as revealed by LMD IASI in the dust belt. After the reanalysis datasets are sampled at IASI overpass times at each grid point for consistency with satellite observations as discussed in section 2.2.5, the annual and seasonal climatology of daytime, nighttime, and day-night differences in DOD from MERRA-2 and EAC4 are presented in Figs. 9 and 10, respectively. Like IASI, the results of the seasonal mean climatology of MERRA-2 and EAC4 DOD from 2008 to 2020 also revealed a higher DOD in

MAM and JJA in comparison with other seasons. The magnitude of the day-night difference in DOD is however very weak in the reanalysis products and largely insignificant as compared to that of IASI (Figs 9. And 10). The magnitude of the day-night difference in MERRA-2 DOD appears to be large only in the Bodélé depression (centered around 17°N, 18°E), with a positive difference throughout the year and a negative difference to the southwest (Fig. 9l-o). Over the northeastern Africa and coastal area of the Arabian Peninsula and central Asia, some areas also show significant negative differences, i.e., with greater

nighttime DOD. The sign of the day-night differences in MERRA-2 DOD is largely consistent with IASI in some parts of the Bodélé depression and southern Arabian Peninsula in DJF and MAM and central Asia in JJA but not in other regions or season.

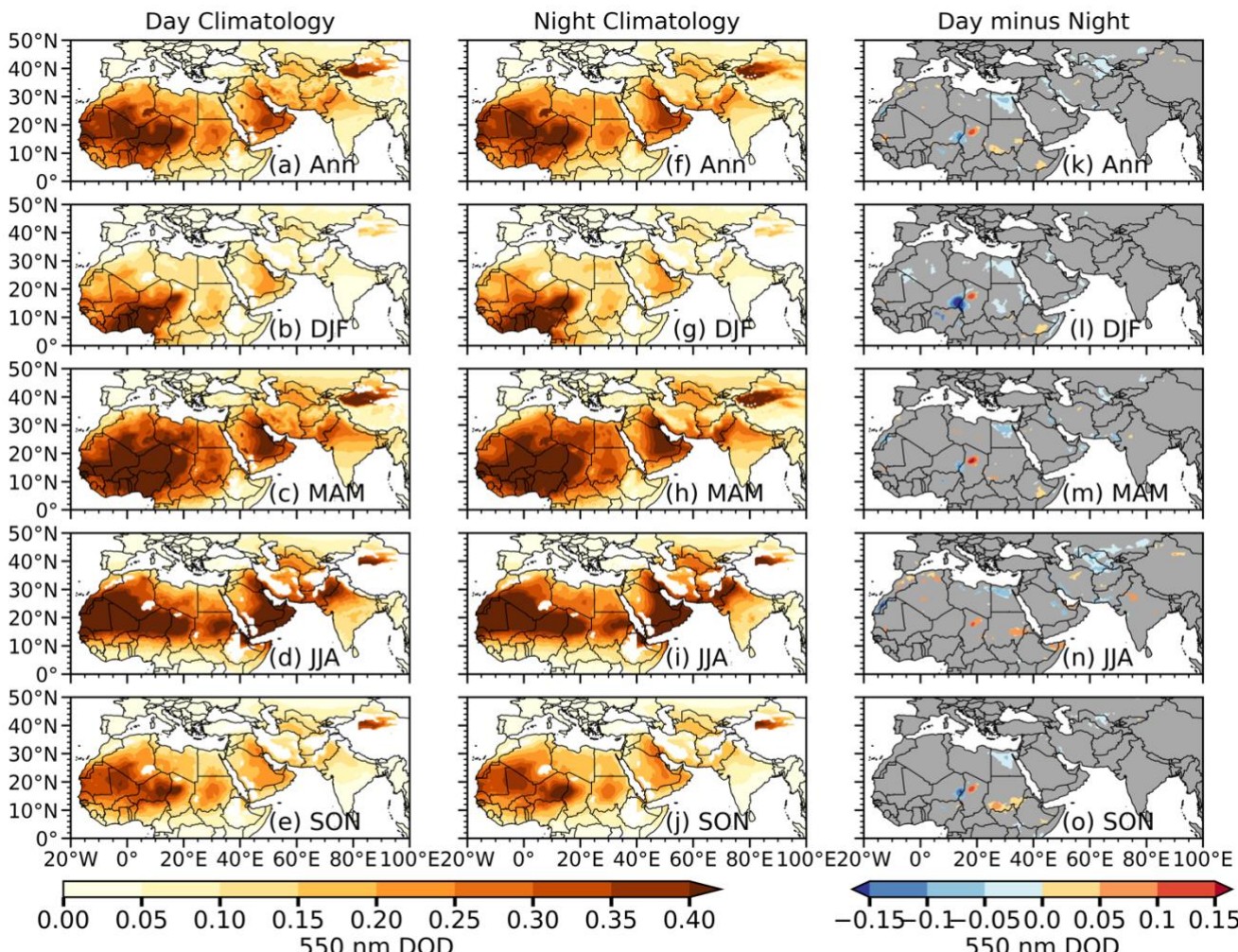

**Figure 9: Annual (Ann) and seasonal means of MERRA-2 DOD averaged from 2008 to 2020 (a)–(e) at daytime (9:30 a.m. local equator crossing time), (f)–(j) nighttime (9:30 p.m. local equator crossing time) based on IASI overpass time at each grid point, and (k)–(o) day-night differences (k–o). White area over land in (a)–(j) denote missing values correspond to IASI DOD. Areas where day-night differences in MERRA-2 DOD do not pass the 95% confidence level (t-test) in (k–o) are masked out in grey.**

A slightly larger portion of the central to northern Sahara, the Middle East, central Asia, and the eastern Taklamakan Desert are characterized by significant and negative day-night differences in DOD in EAC4 (Fig. 10l-o). In most of these areas,

the day-night differences are opposite to that of IASI, except over the northeastern Sahara, the southern Arabian Peninsula, northwestern Sudan in DJF, and central Asia in JJA. In short, aerosol reanalyses in general have difficulties in capturing the day-night differences in DOD shown by IASI. This may be partially because reanalyses do not assimilate nighttime observations (e.g., AERONET lunar data or infrared satellite products) to constrain AOD or DOD.

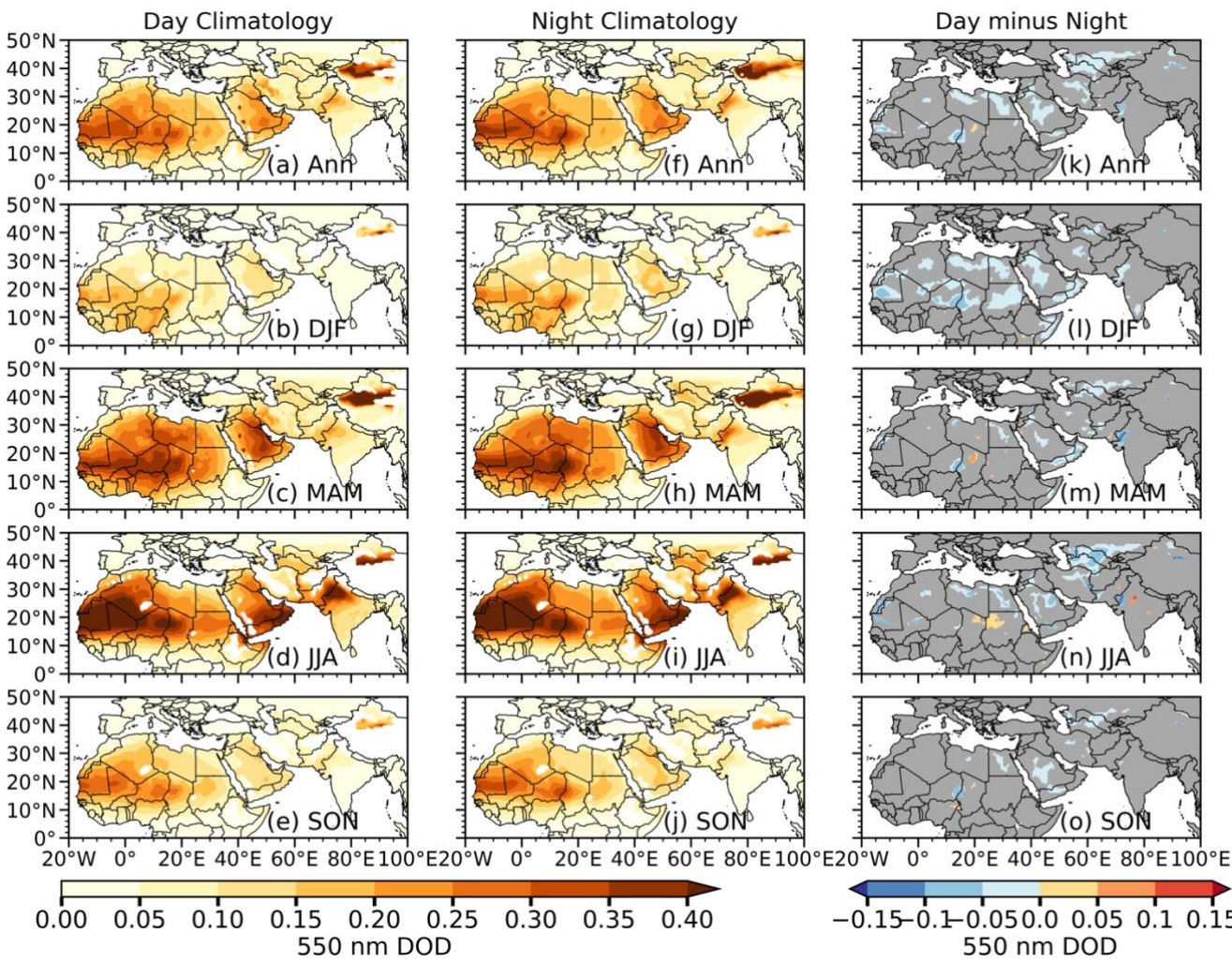

**Figure 10: Same as Fig. 9., but for EAC4.**

### 3.6 Meteorological factors contributing to the observed day-night differences in dust aerosols

In this section we examine the impacts of meteorological conditions on the daytime and nighttime variations in dust aerosols, mainly DOD and dust layer height from IASI, over the dust belt using meteorological variables from MERRA-2, ERA5, IMERG, and LISA observational datasets.

### 3.6.1 Day-night differences in surface winds

Wind speed of appreciable magnitude can enhance dust emissions over dust source regions (Fernandez-Partagas et al., 1986; Todd et al., 2008; Schepanski et al., 2009; Marsham et al., 2011; Fiedler et al., 2013). Wind driven dust emissions over source regions can be suspended in the atmosphere for several hours before depositing onto the surface, thus surface winds early or later than the passage of the IASI instrument can influence dust emissions at IASI overpass time. To understand the impact of surface winds on the daytime and nighttime variations in DOD, we sampled ERA5 surface winds corresponding to IASI overpass times (i.e., 9:30 a.m. and 9:30 p.m. local solar ECT for daytime and nighttime, respectively; Fig. 11), 3 hours (6:30 a.m. and 6:30 p.m. local solar ECT; Fig. S4), and 6 hours (3:30 a.m. and 3:30 p.m. local solar ECT; Fig. S5) prior to IASI overpass time. Daytime wind speed is strong in magnitude and mostly northeasterly over a large area of North Africa in DJF, MAM, and SON (Fig. 11b, c, e) with a maximum in DJF over the central Sahara around the Bodélé Depression in Chad. This is consistent with the findings of Fiedler et al. (2013) and Schepanski et al. (2009) who showed a high frequency of nocturnal low-level jets over the Bodélé Depression in DJF . The strong surface winds over dust source regions, such as the Sahara Desert and the Bodélé Depression, not only favor local dust emissions but also transport dust southward to the Guinea coast (Figs. 4b, c, e). During JJA, following the development of the West African monsoon and Indian summer monsoon, surface winds become southwesterly over the Sahel and the Guinea Coast and over large parts of the Indian subcontinent (Fig. 11d, i). Consequently, high magnitude of DOD is largely located over the northern Sahel and southern central Sahara between 15 °N and 30 °N in North Africa and central to northern Pakistan in JJA (Fig. 4d).

Nighttime wind speeds are slightly weaker in comparison to the daytime (Fig. 11f–j). The magnitude of the day-night difference in surface wind is relatively strong during DJF–JJA, with a maximum in JJA (Figs. 11l–o). In North Africa, the day-night difference in surface wind speed is positive, i.e., with stronger daytime winds, and significant everywhere except over the northern portion of the Sahara along the coast of the Mediterranean Sea where the differences remain negative for all seasons and over the Guinea coast where the differences are negative in DJF, MAM, and SON (Fig. 11l–o). Daytime surface wind speeds are more than 2 m s$^{-1}$ higher than nighttime winds in some areas over the Sahara Desert, likely resulting in stronger dust emissions and higher DOD in the Sahara during daytime. The weaker daytime winds over the central Arabian Peninsula and the Taklamakan Desert indicate that the observed day-night differences in surface winds likely cannot explain the positive day-night differences in IASI DOD in these source regions. Surface wind speed from MERRA-2 (not shown) revealed similar results except that the magnitude of the day-night difference is higher in DJF in MERRA-2. Similar patterns of daytime, nighttime, and day-night differences in surface winds are found at 3 to 6 hours prior to IASI overpass time, but with smaller day-night differences than at IASI overpass time (Figs. S4-5).

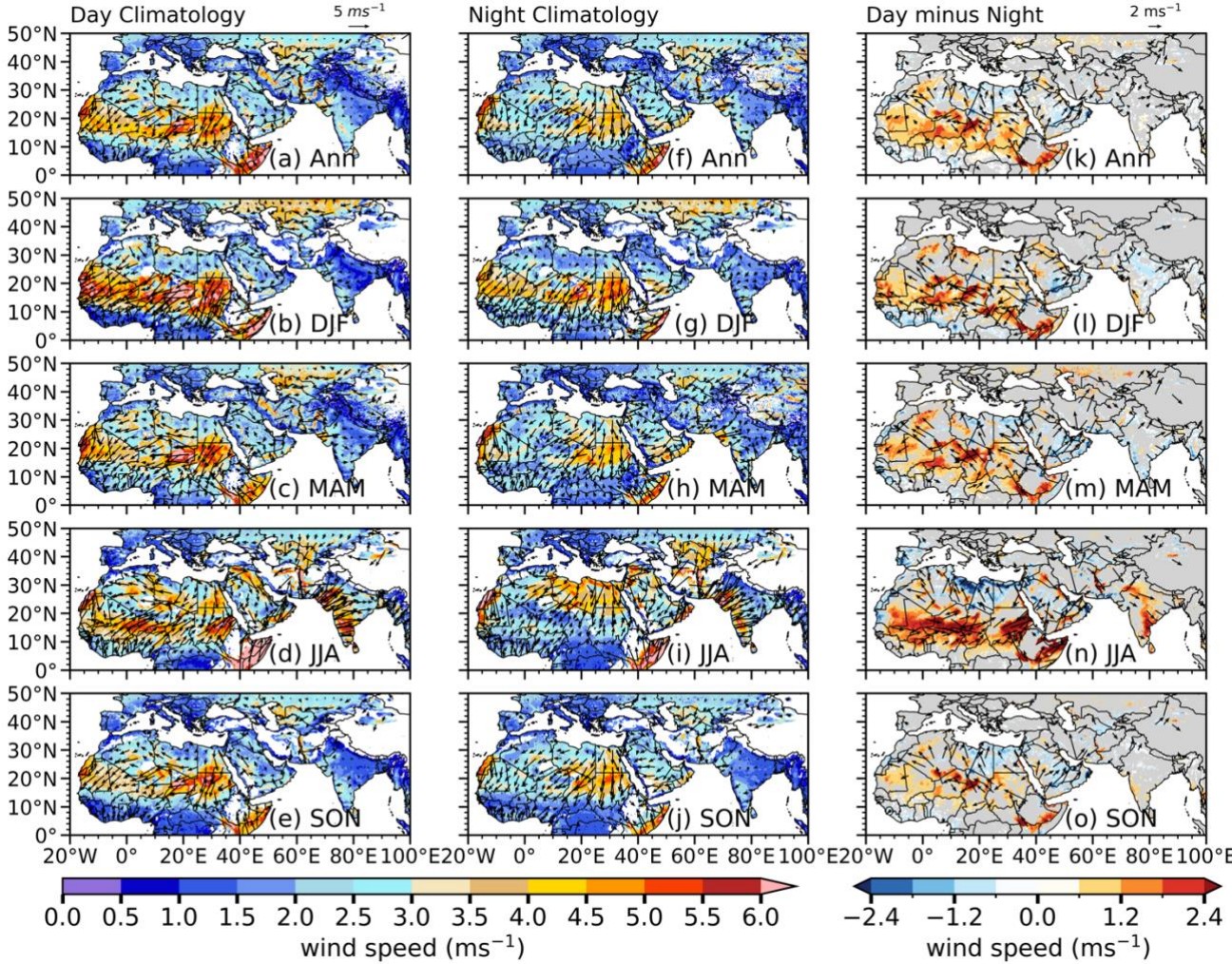

**Figure 11: Annual (Ann) and seasonal mean climatology (2008–2020) at (a)–(e) daytime, (e)–(h) nighttime and (k)–(o) day-night difference in surface winds from ERA5 (unit: m s⁻¹). Data at each grid point are resampled according to IASI overpass time, i.e., 9:30 a.m. local solar ECT during the daytime and 9:30 p.m. local solar ECT at nighttime. Shading shows wind speed, and vectors denote wind directions. Areas where day-night differences in wind speed do not pass the 95% confidence level (t-test) in (k–o) are masked out in grey. Only differences in wind vectors significant at the 95% confidence level are shown.**

While surface winds can affect both the emissions and transport of dust from source regions, the dust uplift potential (DUP; Marsham et al., 2011) better quantifies the dust emission power of winds. Figure 12 shows the climatology of daytime, nighttime, and day-night differences in DUP calculated using surface wind speed from ERA5 reanalysis and a monthly 2D threshold velocity of wind erosion retrieved by Pu et al. (2020). The seasonal climatology of DUP reveals that wind speed capable of dust emissions is predominantly in the northern part of the Sahel to the central Sahara and the central to eastern

Arabian Peninsula, and around the Taklamakan Desert with the strongest DUP in DJF, MAM, and JJA. The day-night difference in DUP is positive and significant in the Sahara, and around the central to eastern Arabian Peninsula, largely consistent with higher daytime DOD in these regions (Fig. 4l–o), indicating stronger daytime dust emissions and likely contribute to the positive day-night differences in DOD (Fig. 4). An attempt has been made to also compare these results with DUP calculated using a constant threshold wind velocity of 7 m s$^{-1}$ following Marsham et al. (2011) and Bergametti et al. (2017), and the overall results are similar except the magnitude of DUP using a constant velocity threshold is slightly less (Fig. S6).

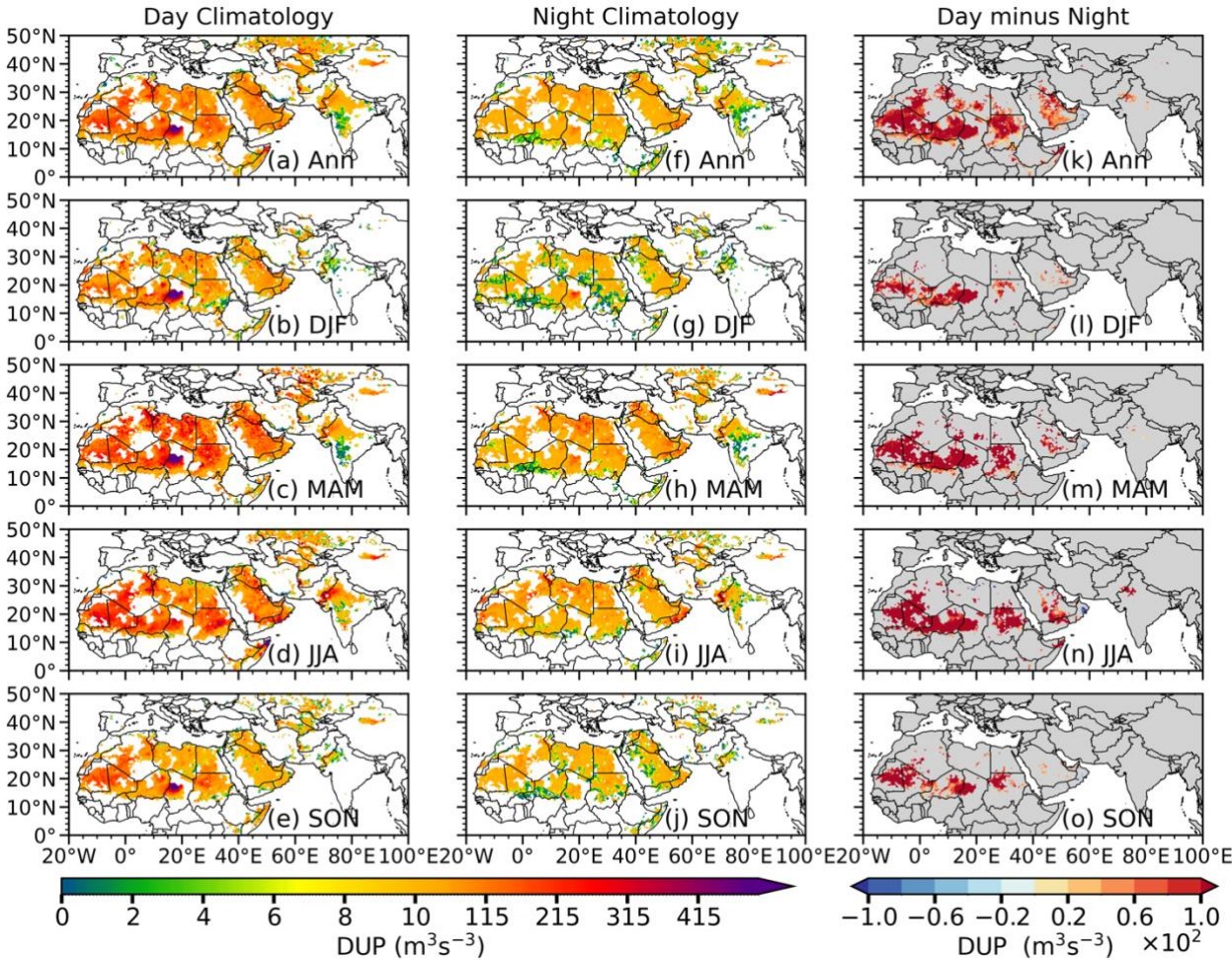

**Figure 12: Annual (Ann) and seasonal mean climatology (2008–2020) of (a)–(e) of dust uplift potential (DUP) at daytime), (e)–(h) nighttime, and (k)–(o) day-night difference using wind velocity threshold estimated by Pu et al. (2020) and surface wind speed from ERA5 (unit: m$^3$ s$^{-3}$). Wind speeds at each grid point are resampled according to IASI overpass time, i.e., 9:30 a.m. local solar ECT during the daytime and 9:30 p.m. local solar ECT at nighttime. Areas where day-night differences in DUP do not pass the 95% confidence level (t-test) in (k–o) are masked out in grey.**

How do diurnal variations in surface winds affect the diurnal cycle of dust aerosols at LISA and AERONET sites? Figure 13 shows the diurnal cycle of observed surface wind speed from LISA station data over LISA sites and ERA5 reanalysis
over AERONET sites. The cyan vertical lines mark the 9:30 a.m. and 9:30 p.m. ECT corresponding to IASI overpass times. Surface wind speeds at LISA sites (Ban, Cin and Mbo) over the Sahel peak in the morning around 10–11 a.m. LST in most of the seasons except at Mbo site in JJA and SON, where surface winds maximize in the afternoon around 3–4 p.m. LST (Figs. 13a–c). A minimum of surface wind speed usually occurs in the evening around 8 p.m. or mid-night, with a secondary minimum around early morning (~ 7 a.m.). The diurnal cycle of surface $PM_{10}$ concentrations show similar maxima in the late morning
around 10–11 a.m. and minima in early morning around 6–7 a.m. (Fig. 8a–c), coinciding with the variations in surface wind speeds (Fig. 13a–c), which is consistent with the findings of Kaly et al. (2015).

The morning peaks (around 7–8 a.m. LST) of surface wind speed at AERONET stations in North Africa (Dak, Iza, SCT, Ilo, and Kof) are consistent with the morning maxima of CAOD (Fig. 8d-h), while the wind speed minima in the early hours (about 9–10 a.m.) at SCT, and late afternoon to evening (around 4–6 p.m.) at most sites (Dak, Iza, Kof, and Ilo; Fig.
13d,f,g,h) are also consistent with the minima of CAOD (Fig. 8d–h). Over the Middle East sites (e.g., Mig, Sha, Mez, and DEW) wind speed generally peaks late in the evening (about 4-6 p.m.) in Mig and Sha during JJA (Fig. 13 i, j) and in Mez and DEW during MAM (Fig. 13 k, l), and also largely coincides with the maxima in CAOD (Fig. 8 i–l). At Mez and Kof stations, the secondary peaks of CAOD in the afternoon or nighttime largely coincide with increases in surface wind speed but not so at other sites such as Ilo. In short, the comparison between the diurnal cycle of surface wind speed and CAOD or $PM_{10}$
concentrations reveal similar diurnal variations, especially for the early morning minima of wind speed and CAOD or $PM_{10}$ concentration and the late morning maxima. Individual sites show some local features depending on their distances to dust sources and ocean, elevation, and seasonality.

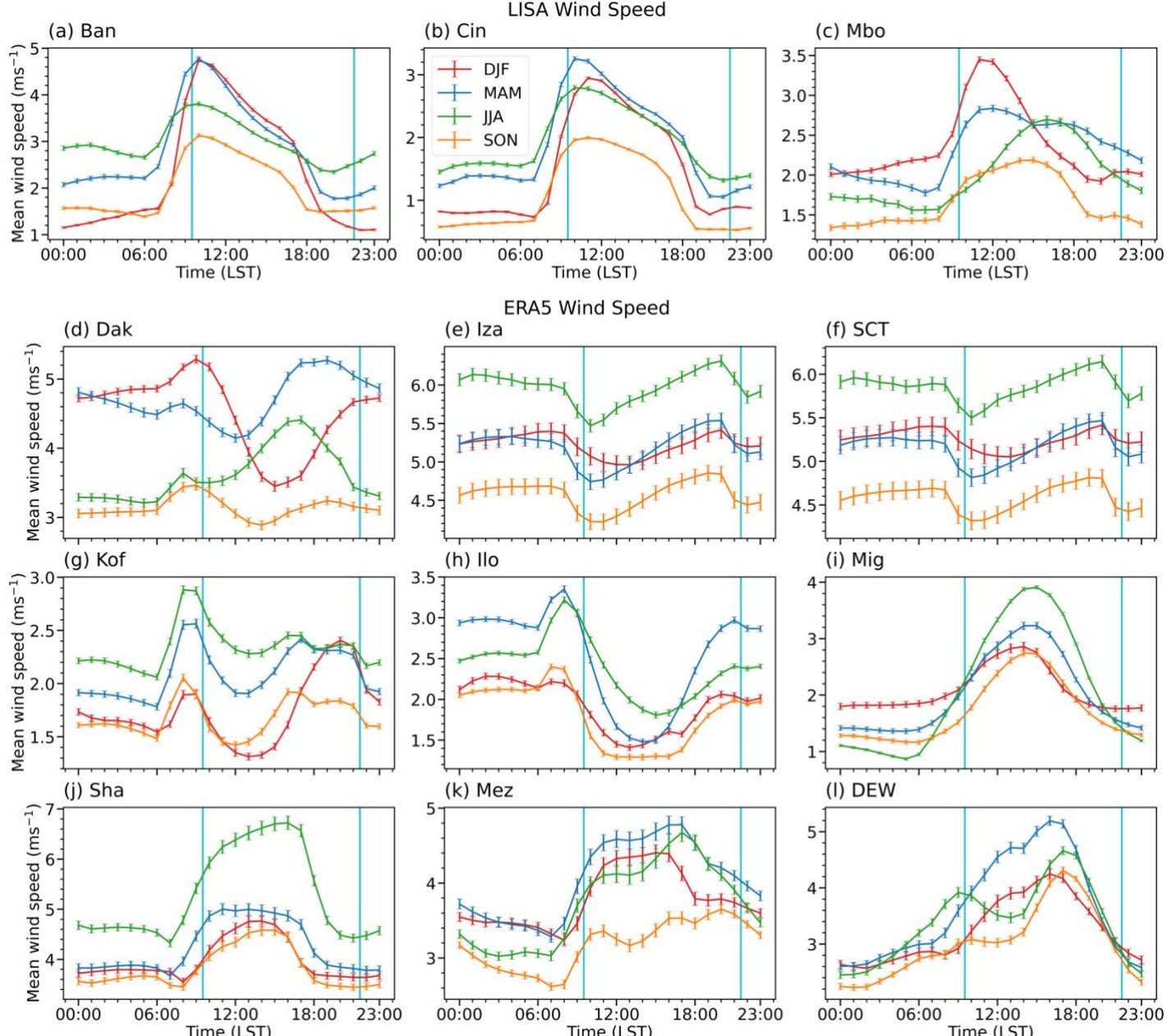

Figure 13: Diurnal cycle of (a)–(c) observed surface wind speed at LISA sites and (d)–(l) ERA5 surface wind speed over AERONET sites in different seasons averaged over 2008–2020 for Ban and Cin, 2008–2012 for Mbo of LISA sites, and over 2008–2020 for ERA5 (unit: m s⁻¹). Note that Iza and SCT sites are very close to each other (see Figs. 1, S1 and Tabels 2, 3), so their surface winds are the same in ERA5. The cyan vertical lines mark the IASI passing time at 9:30 a.m. and 9:30 p.m. ECT. Error bars show standard errors.

### 3.6.2 Precipitation

Precipitation is another factor that can influence the spatiotemporal variability of dust over the dust belt (Engelstaedter et al., 2006; Engelstaedter and Washington, 2007; Knippertz and Todd, 2012; Pu and Ginoux, 2018). Precipitation affects dust aerosols through wet deposition and increased soil moisture that modifies the threshold wind velocity for dust emissions. It is
therefore expected that precipitation events several hours before IASI passage may have impacts on dust emissions at IASI overpass time. To examine the potential impacts of previous precipitation events on the daytime and nighttime variations in DOD, we analyse the annual and seasonal mean climatology of daytime and nighttime precipitation from IMERG sampled at IASI overpass time (9:30 a.m. and 9:30 p.m. local solar ECT; Fig. 14), 3 (6:30 a.m. and 6:30 p.m. local solar ECT; Fig. S7), and 6 (3:30 a.m. and 3:30 p.m. local solar ECT; Fig. S8). Figure 14 shows daytime and nighttime climatology of precipitation
at IASI overpass time. There is low precipitation over large areas of the domain, except the western Guinea Coast in MAM, JJA, and SON, and part of the horn of Africa in MAM. The day-night differences in precipitation are only significant over few spots over the central to the northern Sahara in JJA, showing slightly higher precipitation rate at nighttime (Fig. 14n), which may suppress dust emissions and partially contribute to higher daytime DOD in these regions. At about 3 hours prior to IASI overpass (6:30 a.m. and 6:30 p.m. local solar ECT; Fig. S7), precipitation rates are much higher at nighttime, with larger values
along the southern Sahel and the Guinea Coast in JJA and SON and over the Indian subcontinent. Precipitation rates are even higher at 6 hours prior to IASI overpass (3:30 a.m. and 3:30 p.m. local solar ECTs; Fig. S8). Similarly, the day-night differences in precipitation at about 3 to 6 hours prior to IASI overpass time also show large insignificant areas (Figs. S7-8), indicating that wet deposition may not be playing any significant role in controlling the observed day-night differences in IASI DOD.

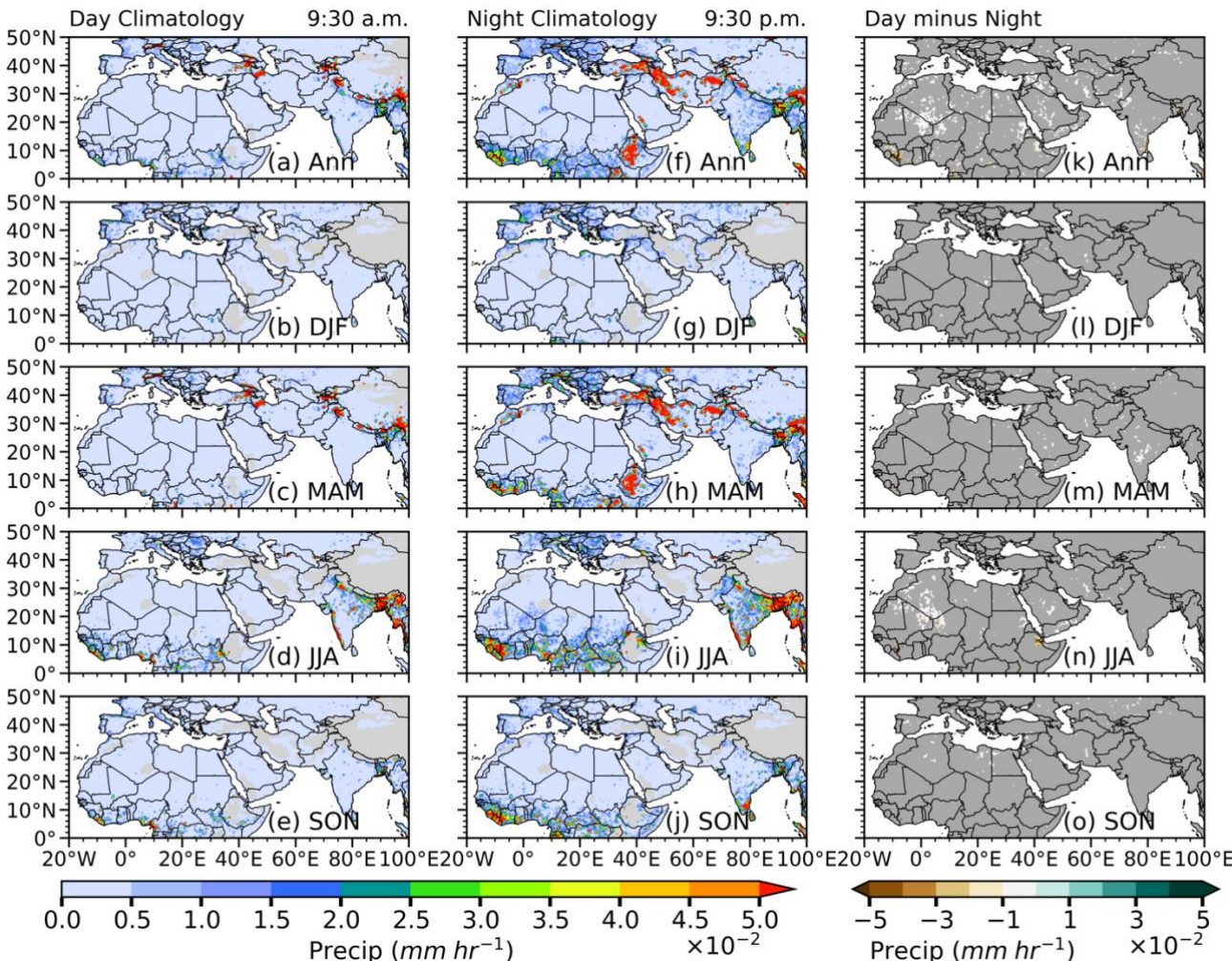

**Figure 14: Annual (Ann) and seasonal mean climatology (2008–2020) of IMERG precipitation (mm hr⁻¹) at (a)–(e) daytime (9:30 a.m. local solar ECT), (f)–(j) nighttime (9:30 p.m. local solar ECT) and (k)–(o) day-night difference. Precipitation at each grid point is resampled according to IASI overpass time, i.e., 9:30 a.m. local solar ECT during the daytime and 9:30 p.m. local solar ECT at nighttime. Areas where day-night differences of precipitation do not pass the 95% confidence level (t-test) in (k–o) are masked out in grey.**


      To further explore the impacts of the diurnal cycle of precipitation on dust aerosols we examined precipitation at LISA and AERONET stations using LISA and IMERG observations (Fig. 15). From LISA observations over the Sahel, precipitation peaks in JJA around early hours of the day (2 a.m. to 8 a.m.) over Ban and Cin in JJA (Fig. 15a, b) and late afternoon to early evening (2 p.m. to 7 p.m.) over Mbo in JJA and SON (Fig. 15c) which is consistent with previous studies in this region

(Marticorena et al., 2010; Kaly et al., 2015). The higher precipitation rates from midnight to early morning in JJA possibly

contributed to the lower daytime $PM_{10}$ concentration in Ban and Cin (Fig. 8a, b) leading to a negative day-night difference in $PM_{10}$ concentration at Ban and Cin (Fig. 4n).

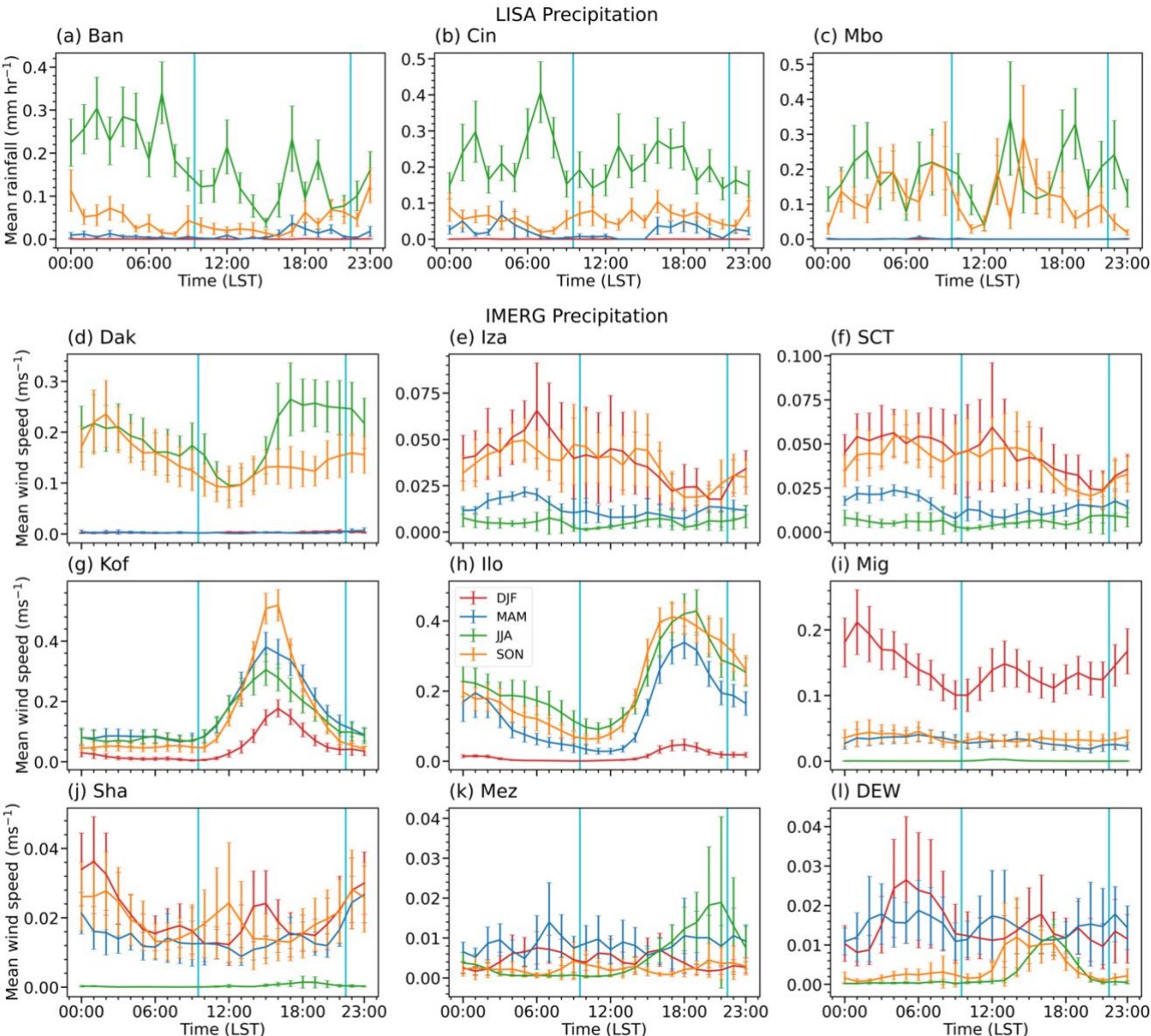

**Figure 15: Diurnal cycle of (a)–(c) observed precipitation over LISA sites and (d)–(i) IMERG precipitation over AERONET sites in different seasons averaged over 2008–2020 for Ban and Cin, 2008–2012 for Mbo of LISA sites, and 2008-2020 for IMERG (unit: mm hr⁻¹). The cyan vertical lines mark IASI passing time at 9:30 a.m. and 9:30 p.m. ECT. Error bars show the standard errors.**

Precipitation from IMERG reveals that Dak, which is collocated with LISA Mbo site, has a peak in JJA around 5 p.m.

to 10 p.m. and around 2 a.m. in SON (Fig. 15d), which is somewhat different from station observations (Fig. 15c). The increase in precipitation in the afternoon and the maxima around 5–6 p.m. (3–4 p.m.) at Dak, Ilo, and Kof sites in JJA and SON (Fig. 15d,g, and h) roughly coincide with CAOD minima around 5–6 p.m. (Fig. 8d, g, and h), suggesting that wet deposition likely reduces airborne dust. Also note that since AEROENT CAOD data are cloud screened, few records are available during precipitation-prone hours. Thus, the scavenging effect of precipitation on dust may not be fully illustrated on the AERONET

CAOD time series. The coastal sites i.e., Iza and SCT have higher precipitation rate in DJF and SON than other seasons, with a maximum at about 6 a.m. in Iza and around dawn in SCT (12–4 a.m.; Fig. 15e, f), and a secondary peak at noon in SCT(Fig. 15e, f). The morning precipitation maxima in Iza and SCT may also contribute to minima in CAOD in the early hours of DJF and SON (Fig. 8e, f). Nonetheless, in the Middle East, the precipitation maxima around 9 p.m. in JJA at Mez (Fig. 15k), 3 a.m. in DJF at Mig and Sha (Fig. 15i and j), and 5 a.m. in DJF at DEW (Fig. 15l) largely correspond to smaller CAOD in a few

hours later in DJF and SON over Sha, Mig, and DEW (Fig. 8i, j, l), but not so evident in Mig and Mez (Fig. i, k).

### 3.6.3 Planetary boundary layer height and atmospheric stability

The planetary boundary layer (PBL) plays a vital role in regulating the vertical mixing and transport of near surface aerosols, including dust aerosols (Knippertz and Todd, 2012). A convective planetary boundary layer on a clear, sunny day over desert regions can enhance dust emissions and vertical transport (Sinclair, 1969; Oke et al., 2007; Ansmann et al., 2009;

Knippertz and Todd, 2012). For regions far away from dust sources with little local emissions, the rising boundary layer likely promote horizontal and vertical dispersal of aerosols, leading to reductions in their concentrations (Petäjä et al., 2016; Pal et al., 2014; Li et al., 2017; Lou et al., 2019). High concentrations of absorbing dust aerosols within the boundary layer can enhance the absorption and scattering of significant amount of solar radiation, decreasing the net radiation at the surface, which can reduce the sensible heat fluxes needed to drive the PBL evolution leading to a much shallow PBL height (PBLH; Li et al.,

2017). A shallower PBLH can further increase surface concentration of aerosols leading to a positive feedback loop (Li et al., 2017). It is thus important to examine the impacts of the PBLH on the day-night differences in dust aerosols.

Figure 16 shows the climatology of PBLH at daytime, nighttime, and the day-night differences over the dust belt from ERA5. The PBLH is highest during JJA at daytime, with higher values over the Guinea Coast, central Sahara, and large areas of Eurasia. In general, the day-night difference in PBLH is positive everywhere in the study domain, with smaller differences

(0~400 m) over major dust source regions, e.g., the Sahara Deserts, the central to eastern Arabian Peninsula, and the Taklamakan Desert (DJF, SON; Fig. 16l, o) but larger differences (>400 m) over the Guinea coast, western Arabian Peninsula, large parts of the Indian subcontinent, and around the Taklamakan Desert (MAM, JJA; Fig. 16m, n). These results are consistent with similar analysis from MERRA-2 (Fig. S9) except MERRA-2 PBLH is much higher than that from ERA5, especially during the nighttime by about 1000~1500 m over the Sahara and the Arabian Peninsula. The discrepancies are largely

due to the different methods used to estimate PBLHs in the reanalyses, with the bulk Richardson number method being used in the ERA5 (Zhou et al., 2021).

A careful examination of these results reveals that the overall pattern of day-night differences in PBLH (Fig. 16k–o) is somewhat similar to the pattern of the day-night differences in dust layer height (Fig. 5k–o) but opposite to the structure of day-night difference in DOD (Fig. 4k–o) in IASI. The larger day-night differences in PBLH over the southern Sahel, the Guinea Coast, and the Indian subcontinent indicate that a growing PBL during daytime is likely entraining dust aerosols into higher altitudes where they are susceptible to upper-level horizontal transport. The dilution may contribute to the negative day-night difference in DOD (i.e., lower daytime DOD than nighttime) in the regions.

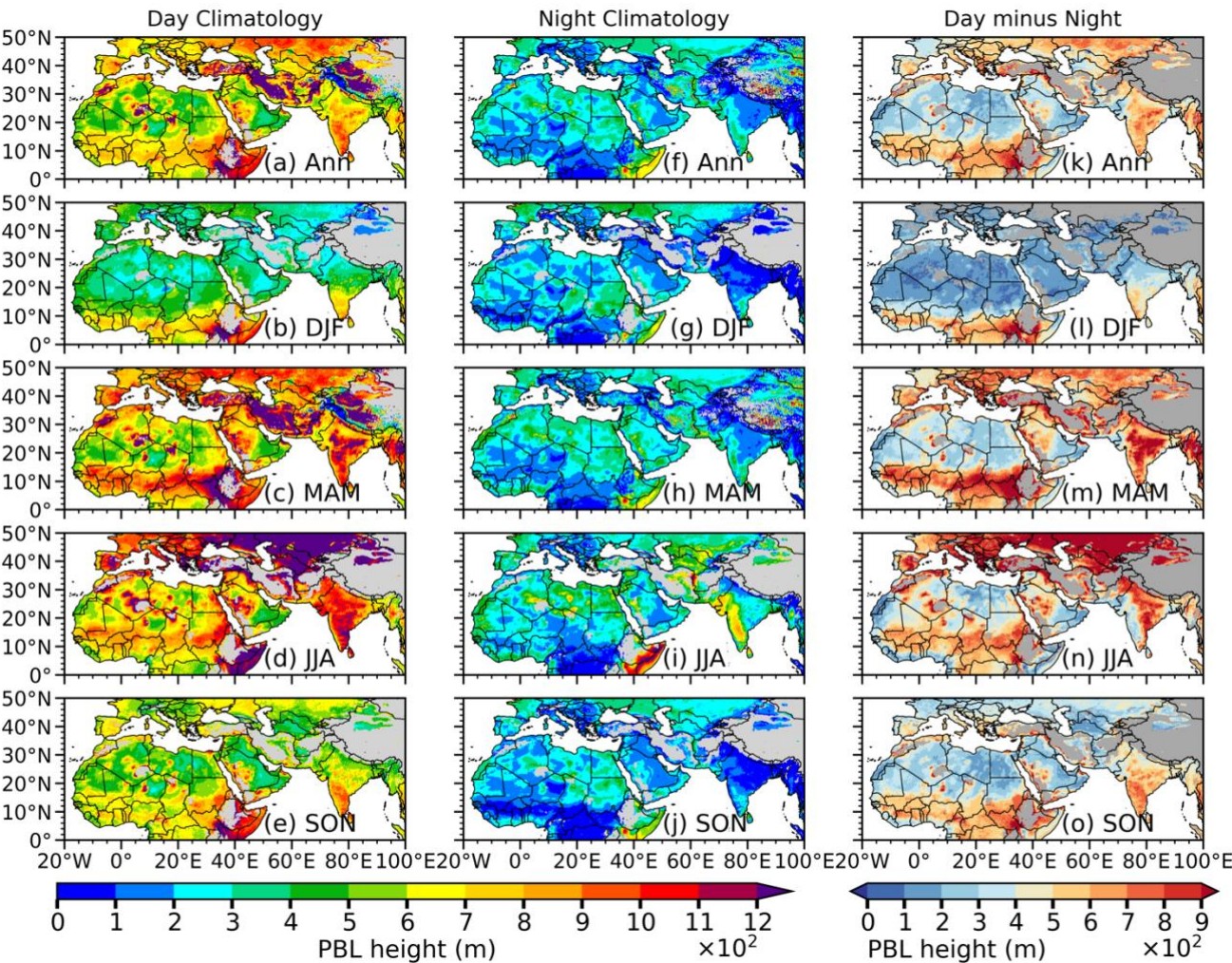

Figure 16: Annual (Ann) and seasonal mean climatology (2008–2020) of planetary boundary layer height (PBLH) at (a)–(e) daytime (9:30 a.m. local solar ECT), (f)–(j) nighttime (9:30 p.m. local solar ECT), and (k)–(o) day-night differences from the ERA5. PBLH at each grid point is resampled according to IASI overpass time, i.e., 9:30 a.m. local solar ECT during the daytime and 9:30 p.m. local solar ECT at nighttime. Areas where day-night differences of PBLH do not pass the 95% confidence level (t-test) in (k–o) are masked out in grey.

An examination of the convective available potential energy (CAPE; Fig. 17) and vertical velocity at 850 hPa (Fig. S10) further show higher CAPE along with rising motion over the Sahel, the Guinea Coast, and the Indian subcontinent during

the daytime (Figs. 17a–e, S10a–e), which may vertically mix dust aerosols into the free troposphere for horizontal dispersion, leading to lower dust concentrations and DOD and higher dust layer heights at daytime. For example, a higher daytime than nighttime CAPE over the southern Sahel (extending to the northern Sahel in JJA and SON; Fig. 17k–o), the Guinea Coast (DJF, MAM; Fig. 17l, m), and the central Indian subcontinent (MAM; Fig. 17m) is consistent with an upward motion in the southern parts of the Sahel, the Guinea Coast, and the central to the northern Indian subcontinent during the daytime (Fig. S10 b–e) and lower daytime DOD (Fig. 4k–o) and higher dust layer height (Fig. 5k–o).

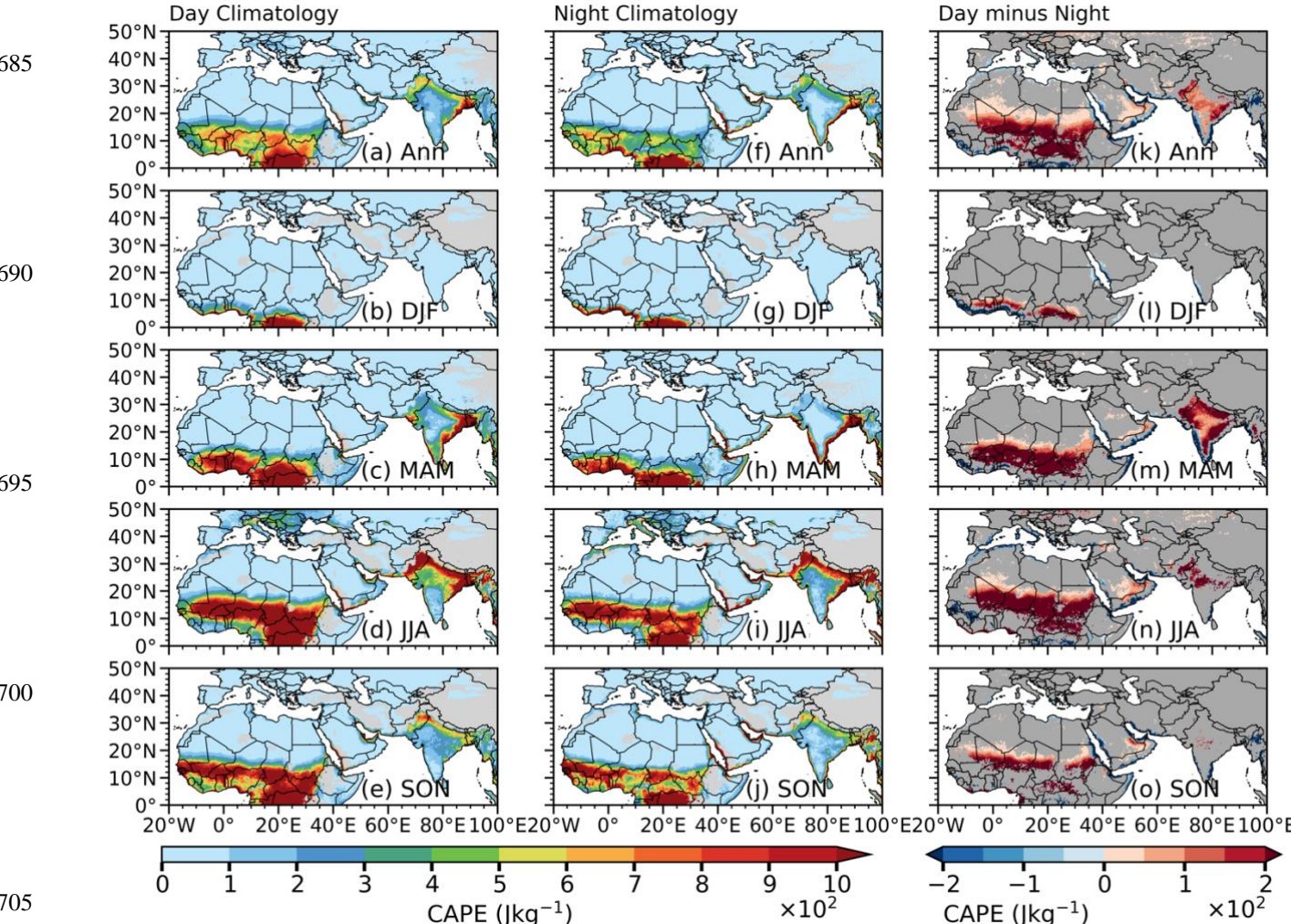

**Figure 17: Annual (Ann) and Seasonal mean climatology (2008–2020) of convective available potential energy (CAPE) at (a)–(e) daytime (9:30 a.m. local solar ECT), (f)–(j) nighttime (9:30 p.m.local solar ECT), and (k)–(o) day-night differences from the ERA5. CAPE at each grid point is resampled according to IASI overpass time, i.e., 9:30 a.m. local solar ECT during the daytime and 9:30 p.m. local solar ECT at nighttime. Areas where day-night differences of CAPE do not pass the 95% confidence level (t-test) in (k–o) are masked out in grey.**

## 4 Discussion

While IASI products provide a viable source of information on global distribution of dust aerosols to complement ground observations, uncertainties associated with the limitations of the instrument, retrieval algorithm, and sampling frequency may add to the uncertainty of our findings. The inability of satellite to observe through clouds is a major setback in aerosol studies using satellite products (Schepanski et al., 2007; Heinold et al., 2013). IASI infrared sensor cannot observe dust aerosols under cloudy convective systems such as haboobs which often occur over North Africa in the evening periods, likely leading to a "morning bias" with more available data at the morning overpass i.e., 9:30 a.m. local solar ECT (Chédin et al., 2020). This could possibly affect the day-night differences in DOD and dust layer height observed from IASI over the convective regions (the Sahel, the Guinea Coast, and the Indian subcontinent) especially on the daily time scale, but likely has little effect on the monthly time scale or climatological mean (Chédin et al., 2020). On the other hand, a rigorous cloud masking method is used for LMD IASI products to ensure high confidence in cloud identification (Pierangelo et al., 2004; Crevoisier et al., 2009; Pernin et al., 2013; Capelle et al., 2018). Due to this, some aerosol loading, especially over the dust source regions, could be mistaken as clouds and screened out, leading to an underestimation of the actual DOD (Capelle et al., 2018).

Other possible sources of uncertainty of IASI retrievals include weak sensitivity to dust aerosols in the first hundred meters above the surface and difficulty in capturing low DOD of similar order or smaller than the sensitivity of the instrument (Capelle et al. 2018). While the passing times of IASI largely coincide with the times of the two most important dust emission mechanisms in the Sahara Desert, i.e., breaking down of the nocturnal low-level jets in the morning and mesoscale convective systems (haboobs) at late afternoon to evening hours (Schepanski et al., 2009; Knippertz and Todd, 2012; Marsham et al., 2013; Chédin et al., 2020), some small dust events before and after the passage of IASI (9:30 a.m. and 9:30 p.m. local solar ECT) could be missed. For those large events that occur a few hours before or after IASI observations, while IASI may still be able to capture them, the location of dust plumes is often shifted from their original locations depending on the direction of the prevailing winds. Future studies of dust aerosols using instruments with different overpassing times likely will complement and improve our understanding of the diurnal cycle of dust aerosols. However, despite these challenges, the day-night differences in IASI DOD are largely consistent with the day-night differences in CAOD and $PM_{10}$ concentrations from ground observations. The presence of orbital gaps around the tropics in current IASI products is partially addressed by the launches of IASI onboard MetOP-B and MetOP-C satellites in 2012 and 2018, respectively (Carboni et al., 2013; Klüser et al., 2013; Chédin et al., 2020). Future investigations using IASI from these satellites and algorithms different from LMD are warranted to confirm and overcome some of the limitations in this study.

We used station products (i.e., AERONET CAOD and LISA $PM_{10}$) to evaluate IASI DOD and examine the diurnal variability of dust as a complement to the day-night differences in IASI DOD. However, many factors may affect the reliability of station data as well, making them less reliable to be used in this study. AERONET products are provided in three levels based on the quality of the data i.e., level 2 (cloud screened and quality controlled), level 1.5 (cloud screened but not quality

assured), and level 1.0 (raw data: neither cloud screened nor quality controlled). A lot of effort was made to use level 2.0 data in this study, but there were sites where level 2.0 data was unavailable, so, level 1.5 data was used instead. Moreover, AERONET lunar data is still under development and level 2.0 is not yet available, hence level 1.5 data was used. The use of

level 1.5 data and generally smaller sample size of lunar data than solar data could also introduce additional uncertainties to the examination of day-night differences in dust aerosols. In addition, comparing between different observational platforms can be challenging. The sensitivity of AERONET observations to cirrus clouds can introduce significant amount of uncertainties in the aerosol retrievals (Smirnov et al., 2018) especially over sites close to the tropics. In addition, comparing between different observational platforms can be challenging. For instance, comparing between mass concentration ($PM_{10}$) and

vertically integrated (DOD or CAOD) quantities is not straight forward as they characterize different aspect of dust. Despite these uncertainties, both station and IASI products largely agree well on the seasonal climatology of dust and in some sites on the day-night differences in dust aerosols over the dust belt.

In addition to the meteorological variables discussed above, we also found slightly higher relative humidity at 750 hPa during nighttime over the Guinea Coast and the south coast of India (not shown) that may partially contribute to the higher

nighttime DOD via the hygroscopic growth of aged dust, although the overall effect is hard to quantify in observations. Land surface variables such as soil moisture may also affect dust emissions in semi-arid regions. However, our examination of soil moisture from ERA5 showed that the difference in soil moisture between IASI daytime (9:30 a.m. local solar ECT) and nighttime (9:30 p.m. local solar ECT) overpasses is small and insignificant, indicating a likely negligible impact on the day-night differences in DOD. While surface wind speed, precipitation, PBLH, and atmospheric stability all affect day-night

differences in DOD and dust layer height to some extent, they may be fundamentally driven by common factors, such as diurnal cycle of surface radiation, and modulated by local land surface and circulation features. Additional sensitivity tests are needed to further quantify the relative contribution of individual factors to the day-night differences in dust aerosols revealed by IASI and station data and will be addressed in our future study.

## 5 Conclusions

While dust aerosol remains one of the key factors affecting the climate system, constraining the full diurnal cycle of dust from current visible satellite products and sparsely located ground observations presents a challenge, which continues to contribute in large portion to the sources of uncertainties in estimating the total radiative forcing of aerosols and projecting climate change. Using the equal quality performance for daytime (9:30 a.m. local solar ECT) and nighttime (9:30 p.m. local solar ECT) observations, and global coverage at fine spectral and spatial resolutions of LMD IASI products, this study

investigates the day-night differences in dust aerosols over the global dust belt of North Africa, the Middle East, and Asia. A comparison between IASI 10 μm (scaled to 500 nm) DOD and AERONET 500 nm CAOD revealed an overall correlation coefficient of ~0.75 for 46 solar sites and ~0.55 for 11 lunar sites, indicating IASI exhibits reasonably well performance in capturing the spatiotemporal variability of dust events over the dust belt.

IASI showed significant (95% confidence level) day-night differences in DOD and dust layer height within the dust belt, with higher DOD and lower dust layer height during the daytime over dust source regions in the central to northern Sahara, the Arabian Peninsula, the northwestern Indian subcontinent, and the Taklamakan Desert. Over the southern Sahel to Guinea Coast and large area of the Indian Subcontinent, nighttime DOD is observed to be higher than daytime, along with lower dust layer height at nighttime. The day-night differences in DOD are larger and more significant in MAM and DJF than other seasons, while day-night differences in dust layer height show little seasonal variations. The higher daytime DOD in dust source regions (e.g., the central Sahara, the Arabian Peninsula, northwestern Indian subcontinent, and Taklamakan Desert) are likely associated with higher dust uplift potential (DUP) during daytime in these regions and larger magnitude of positive day-night differences in surface wind speeds in the Sahara Desert. Over some spots of the Sahara, the central Arabian Peninsula, and the northwestern Indian subcontinent, slightly higher nighttime precipitation rate may reduce airborne dust and partially contribute to higher daytime DOD in JJA as well.

The low daytime DOD over downwind regions, such as the southern Sahel, Guinea Coast, and the central to southern Indian Subcontinent, coincides with a relatively higher planetary boundary layer height (PBLH) and greater convective available potential energy (CAPE) at daytime that corresponds to an unstable atmosphere.  The growing PBLH during the daytime likely entrains dust aerosols into upper levels, resulting in a higher dust layer height and favouring horizontal transport of dust, which likely dilutes column concentrations of dust and results in lower DOD during daytime.

Seasonal analysis of day-night differences in DOD from MERRA-2 and EAC4 revealed that reanalysis products largely capture the temporal and spatial variability of DOD on the seasonal scale but failed to capture the day-night differences in DOD in large parts of the dust belt except in a few dust hotspots over North Africa, such as the northeastern Bodélé Depression in DJF and MAM (MERRA-2), and over parts of northeastern North Africa in DJF, JJA and SON and over the southern Arabian Peninsula in DJF (MERRA-2 and EAC4).

Using ground-based measurements from LISA and AERONET observations, we have shown that dust aerosols exhibit a spatially varying diurnal cycle across the dust belt with higher coarse-mode aerosol optical depth (CAOD) and $PM_{10}$ concentrations in the morning hours (7–9 a.m. in CAOD and 9–11 a.m. in $PM_{10}$) and late afternoon (3–4 p.m. in CAOD and 6–9 p.m. in $PM_{10}$) and midnight ($PM_{10}$) to early morning (CAOD) in the Sahel, higher CAOD  in the afternoon (3–4 p.m.) and early morning (2–5 a.m.) in the Arabian Peninsula. The day-night differences in CAOD between 9:30 a.m. local solar time (LST) and 9:30 p.m. LST are also largely consistent with day-night differences in IASI DOD in sign and magnitude at some sites but not others, possibly due to a smaller sample size of AERONET lunar data.

In conclusion, this work has shown that daytime dust aerosols around 9:30 a.m. local solar ECT over the dust belt is significantly different from nighttime at 9:30 p.m., local solar ECT, and such day-night differences are largely influenced by the local meteorological conditions, primarily, surface circulation, precipitation, and turbulent motion over the dust belt. Despite the uncertainties associated with satellite products and station data, the findings add to our current understanding of the diurnal of cycle of dust in major dust source and downwind regions.

**Data availability**

Data used in this study can be downloaded from the links shown in Table 1.

**Code availability**

Analysis codes can be provided by the corresponding authors upon request.

**Author contributions**

BP and QJ conceived the study. JZT performed the analysis under the guidance of BP and QJ. JZT wrote the paper with input from BP and QJ.

**Competing interest**

The authors declare that they have no conflict of interest.

**Acknowledgement**

The authors would like to thank David Mechem and David Rahn for their helpful discussion and valuable suggestions on this paper. We also thank Brian Harr for helpful edits. IASI is a joint mission of EUMETSAT and the Centre National d'Etudes Spatiales (CNES, France). The authors acknowledge the AERIS data infrastructure for providing access to the IASI data in this
study and CNRS-LMD for the development of the retrieval algorithms. The helpful and constructive comments from the two anonymous reviewers greatly improved the paper and are sincerely appreciated.

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
