# Peer review of "Understanding day-night differences in dust aerosols over the dust belt of North Africa, the Middle East, and Asia"

_Atmospheric Chemistry and Physics, 2022_

## Author Comment (AC1)

**Responses to Reviewers' Comments**

We would like to express our sincere gratitude and appreciations to the reviewers for their valuable and constructive suggestions. The specific comments and concerns of each reviewer is addressed below. Line numbers refer to the lines in the manuscript with tracked changes.

**Reviewer #1 Comments:**

I understand your study is a very comprehensive one. Yet, if possible, it would be nice if you could shorten the abstract a bit. Just a suggestion to consider. (e.g., line 24 describes DOD over the 'downwind region'. Perhaps mentioning this information in the abstract would be important to someone who knows which 'downwind region' you are referring to. For a general reader, this line in the abstract doesn't provide much insight)

Thank you very much for pointing this out. We have shortened the abstract. We have also replaced the "downwind region" with "Sahel and the Indian Subcontinent (in line 25)" to avoid confusion.

line 75: Perhaps adding local time would be helpful here.

We have added "about 5:00 a.m.–10:00 a.m. local solar time at the western and eastern boundaries of the Sahara" to line 71.

line 135: FIG.1 Site labels are not clear even after zooming in. I think you may have to include higher-resolution figures. Where is 'Mez'? (I see something that looks like 'Mer'. Is it a typo?).

We have updated the figure to increase the clarity of the labels on Fig. 1.

One main concern I have is the very specific method you've used to determine the dust layer altitude. What would happen if you chose the mean (or median) altitude instead of the 50th-percentile altitude (line 170)? Why don't you use a consistent approach to determine the layer altitude from CALIOP? You used the mean of the highest and lowest layers to determine the altitude from CALIOP. Perhaps the highest or lowest detected layers could be optically thin and the deduced layer altitude may not serves your intentions.

Thank you very much for the comment. In lines 182-183, we described the dust layer height defined by the IASI dataset: it is the altitude at which half of the DOD is above and the other half below (Peyridieu et al. 2013; Capelle et al. 2014). It is considered as the "infrared optical equivalent to the centroid of the real vertical profile" (Chédin et. al. 2020). We don't have the information to calculate the mean or median altitude of IASI dust plume height. There is no straightforward method for deriving dust layer height from CALIOP. The dust layer altitude from CALIOP L3 data is reported in percentiles, with the first percentile (10%) containing the minimum base altitude and the last percentile (90%) containing the maximum base altitude. We averaged the two to obtain CALIOP dust layer height. The uncertainties due to different definitions of dust layer heights between IASI and CALIOP have been extensively examined by Kylling et al. (2018), who calculated CALIOP dust layer height using both arithmetic mean (similar to ours) and cumulative

extinction height (as IASI) and found that the RMSE between the two methods is ~0.625 km. We updated our discussion in lines 428-433 to better clarify this: "The differences between IASI and CALIOP dust layer height could be attributed to several factors (Peyridieu et al., 2013; Chédin et al., 2020), such as different definitions of dust layer height, e.g., arithmetic mean dust layer height in CALIOP versus cumulative extinction height in IASI, and different overpass times of the two instruments (CALIOP lags IASI in about 4 hours). Kylling et al. (2018) found that the bias of dust layer height in IASI (LMD version) would be lower if CALIOP dust layer height was defined by cumulative extinction height instead of arithmetic mean and was shifted to the observation time of IASI. Their results (their Table 3) show a difference of  $-0.053\pm1.339$  km between LMD IASI and CALIOP for the cumulative extinction and  $-0.607\pm1.187$  km for arithmetic mean. "

line  $\sim$ 700: The discussion covers most of the potential uncertainties. If I'm not mistaken, you haven't mentioned the biases due to the use of coarse AOD as dust optical depth.

Thanks for the comment. In lines 208-209 we discussed the uncertainties associate with the coarse mode AOD (CAOD) from AERONET data, including the fact that CAOD may contain other coarse particles such as sea salt at coastal sites. Later when comparing the AERONET CAOD and IASI DOD we also discussed potential influences of sea salt that may bias CAOD (e.g., lines 384; 470). We added a paragraph in lines 741-754 to include uncertainties in using CAOD and PM10 concentrations to approximate dust.

**Reviewer #2 Comments:**

**Overall major comments:**

1. "Dust activities" in the title and a bit everywhere in the manuscript: what does that mean precisely? My first thought was that it meant emissions / transport / deposition (the dust cycle – and I would then write in singular "dust activity"), but then mostly AOD/DOD and layer height are discussed, not the dust cycle. I think it would be more correct to replace dust activities by dust optical depth and layer height.

Thank you for pointing this out. "Dust activities" to refers to the day-night variations in IASI DOD and dust layer height, along with the variability revealed by station data such as AERONET CAOD and LISA  $PM_{10}$  concentrations. We replaced "dust activities" with "dust aerosols" to avoid confusions in the paper.

2. Four IASI dust retrieval algorithms exist, all publicly available through the climate data store (cds.climate.copernicus.eu), directly as level 3 gridded products and containing both the 10µm DOD and the converted 550nm DOD (very close to the 500nm value) following the best method defined by the data providers. Although I do not contest at all the use of the LMD product, maybe it should be made clearer why (only) this product was used in the analysis (and not just because it has good validation, at least 2 other products do also have good validation, check 10.1029/2018JD029701 and 10.5194/amt-12-3673-2019 for instance). In addition, it needs to be clear, from the beginning and all along, which IASI dust product is used (here it becomes clear only at section 2.2.1 but an annual IASI DOD map has been presented earlier in the manuscript). I strongly recommend using something like LMD IASI DOD instead of IASI DOD. Because different conclusions might arise if using different data sets (and that would actually be a very interesting comparison).

Thank you very much for the suggestion. Indeed, there are at least four retrieval methods (IMARS, LMD, MAPIR, and ULB) of IASI DOD, and level 3 DOD and dust layer height (not for UBL) on a  $1^{\circ}\times1^{\circ}$  resolution are openly available on the climate data store (cds.climate.copernicus.eu). Some products have been extensively validated (e.g., Callewaert et al. 2019; Clarisse et al. 2019) as the reviewer pointed out. We updated lines 166 to 173 to better explain why LMD DOD and dust layer height are used in our study: "While several retrieval methods are available for IASI DOD and dust layer height (e.g., Klüser et al. 2011; Callewaert et al. 2019; Clarisse et al. 2019), we use the version from LMD (Peyridieu et al., 2013; Capelle et al., 2018) as it provides global retrievals of both DOD and dust layer height. LMD IASI dust products show good consistency with ground observations (Capelle et al., 2014, 2018; Peyridieu et al., 2013; Zheng et al., 2022) and well performance in comparison with other IASI DOD datasets (Klüser et al. 2015). The publicly available L2 data also allow us to validate and compare with ground observations in our study domain and time period and to interpolate the data to a reasonably high spatial resolution (i.e.,  $0.5^{\circ}\times0.5^{\circ}$ ) to facilitate our study."

We also updated the whole manuscript to emphasize that we used LMD version of IASI DOD and dust layer height in this study. We totally agree that IASI DOD and dust layer height from different algorithms may generate slightly different results and mentioned in Section 4 (Discussion) that "...Future investigations using IASI from these satellites and algorithms different from LMD are warranted to overcome some of the limitations in this study." (lines 745-746).

3. The conversion between thermal infrared and visible DOD is very confusing and not consistent all along. Figure 1 shows IASI scaled to 500nm using an average ratio from all AERONET stations, while the validation uses a different ratio at each AERONET station, and even a (much) different ratio for solar and lunar observations. This means that the data used overall is not the same as the validated data, and that even at the same location, the conversion is done differently for day and night time data. Then from section 3.2 the AERONET data is now scaled to 10µm using the mean scaling factor (solar +lunar), so again not what was validated. Lines 350-351 do mention that "using individual ratios will slightly improve the consistency between AERONET CAOD and IASI DOD but may lead to some biases in the day-night differences". I agree to that, but then why for the validation a different ratio is used for each station, and even for solar and lunar at the same station? Finally in Figure 7 no conversion at all is done and the plots show IASI DOD at 10µm and AERONET CAOD at 500nm, making it more difficult for the reader to compare values. My advice on this is to convert only once either to 500nm or to 10µm with, I guess, a constant value. Then use the data for the validation and all analyses. Even better would be to use the already converted IASI DOD at 550nm from the climate data store because then it is done according to what the data provider identified as the best way. This will most probably not change conclusions significantly but will make the methodology more robust / logical / understandable.

Thank you so much for raising this concern. We have made several changes to avoid such confusions. With a proper collocation between LMD IASI DOD and AERONET observations in this revision, the average IR/VIS ratios for solar and lunar sites are 0.62 and 0.57, respectively (see revised Tables 2 and 3). We therefore used a constant IR/VIS ratio of 0.6, obtained by averaging between 0.62 and 0.57 to scale IASI 10  $\mu$ m DOD to visible (500 nm) equivalent DOD to compare with AERONET CAOD. The same ratio is now applied to IASI DOD in both validation and analysis. We have modified lines 277-279 to clarify. We also updated Figs. 4, 6, 7, and 8 to scale IASI 10  $\mu$ m DOD to visible (500 nm) equivalent.

4. The local solar time of IASI observations as usually reported (about 9h30 and 21h30 equator crossing) is here considered as the actual truth for all IASI observations. I want to clarify that this crossing time is for the satellite / platform (or pure nadir observations). It does not mean that all observations occur at that precise local solar time, considering the large swath of IASI and also (less important here) that an orbit takes about 100 minutes. First of all, it is an approximate time at the equator. Second for example at 40°N the crossing times would be about 11 minutes earlier / later (100 minutes for 360° latitude). Finally and most importantly: the swath of 2200km (1100km of each side) means about 10° longitude if close to the equator (the orbit is polar, not exactly N to S but very close to

it, so the swath is in the E-W direction) or  $13^{\circ}$  longitude at about 40°N latitude. For local solar time, this means respectively about 40 or 52 minutes local time difference between the extremities of the scan line and the pure nadir position, on each side. This might sound not so important, but it is when trying to compare IASI with local observations close to dust emission peak times. At the end, in an average of all IASI observations close to the Equator, one will have data from about 8:50 to about 10:10 (same pm) at the equator (and +/- 25 minutes at 40° latitude for example). This must be taken into account in all discussions, and also in selecting the data to be compared. (If needed, the real observation time of IASI is reported with the data, for each observation, in UTC time). It is already partly covered by taking an average from 9:00 to 10:00, but I think this average needs to be extended by at least a half hour earlier and later.

Thank you for the detailed explanation and suggestions. We revised the comparison between IASI and all the observation datasets and reanalysis variables (Figs. 2–4, 6–7, 9–12, 14, 16, 17). To avoid the problems you pointed out above, we collocated the observational/reanalysis datasets with IASI overpass time (UTC) at each grid point using level 2 IASI products. See details in lines 260-265. For ground observations (e.g., AERONET, LISA) we sampled station data within  $\pm 30$  minutes of IASI overpass time and IASI pixels within a 30 km radius of stations.

5. Validation against AERONET: I am very sceptical in the way this validation is done, both because of the time averaging of AERONET around expected IASI observation time (instead of using real time co-locations based on IASI exact observing time) and because of the station-dependent conversion factor, even more because those are different for day and night.

Thanks for the comment. The reviewer was concerned about the collocation of AERONET CAOD and IASI DOD and station-dependent IR/VIS conversion factor. We modified lines 262 to 263; 278 to 279 and updated Figs. 2–3 to address these concerns. AERONET station data are sampled within  $\pm 30$  minutes of IASI overpass time based on the time information of pixels in L2 data, while IASI pixels are averaged within a 30 km radius of the AERONET stations. We also converted IASI DOD at 10  $\mu$ m to its 500 nm equivalent using a uniform IR/VIS factor, 0.6 for all the sites.

6. A large effort is given in validating the IASI DOD and some information is provided for IASI dust layer height but no information is provided on the expected quality of other data sets used in this study.

Section 2.2 describes the various data sets used in this study, and the quality of the datasets used in this work are also discussed when such information are available, e.g., lines 254-256 for CALIOP, lines 281-284 for AERONET, lines 380-384 for IMERG-GPM. We have however dedicated more effort in describing the uncertainties associated with IASI because it is the primary data used in our study. LMD IASI DOD has been validated by Peyridieu et al. (2013) and Capelle et al. (2014; 2018) against AERONET AOD or CAOD, but only for daytime and over shorter time periods, e.g., 2007 up to 2016. Here we extended

previous studies to examine both daytime and nighttime IASI DOD in our study domain over 2008–2020.

7. To study the impact of the winds / precipitation on DOD, one should not just look at those values at the moment of the IASI overpass (not forgetting it is not precisely 9:30) but also some time before. Indeed, for example if there was a strong wind/emission a few hours before the IASI overpass, the emitted dust would most probably remain in the air, at least partly. In addition, a larger mean wind speed is not enough to explain more emissions, and mean wind speed is not the right value to look at: a relatively low mean value could hide some very strong winds on specific days, while most days only present weaker winds. For precipitation: it will reduce the DOD for some time afterwards (due to wet scavenging, therefore removing dust from the air mass), and inhibit dust emission also for some time afterwards – depending on the amount of rain and how quickly the surface dries. It makes no sense to look at precipitation precisely at the IASI overpass times. Lines 608-610 mention an attempted analysis looking at precipitation two hours prior to IASI observations. This is a start, but again why specifically 2 hours prior to IASI observations? What is needed is to look during a few hours before IASI observations (if any precipitation took place during a time frame). These 2 sections need to be completely reworked.

We appreciate all these suggestions from the reviewers. We totally agree that the influence of a strong winds on dust plume may be persistent and averaging surface wind speeds will smooth out strong winds. The analysis of surface winds (speed, direction) at IASI overpass time helps us to understand how climatologically wind patterns may affect the distribution of DOD. In the update Fig. 11, the wind speed/direction at each grid point is resampled according to IASI overpass time to facilitate comparisons. While the climatologically mean of surface winds can smooth out some high-speed winds, they are consistent with the climatologically mean of DOD. To clarify the potential persistent impacts of wind on DOD, we analyzed surface wind speed at IASI overpass time (Fig. 11), 3 and 6 hours prior to IASI overpass time (Figs. S4–5). Lines 541, 560 -563 are added to discuss this. Overall, the day-night differences in surface winds speed are the strongest at the IASI overpass time. We also realize that not all the surface winds contribute to dust emissions, so dust uplift potential (DUP; Fig. 12) is examined, and it shows stronger DUP over the Sahara during daytime than nighttime, which is more consistent with the pattern of the day-night differences in DOD than surface wind speeds.

Similarly, we examine precipitation at IASI overpass time, 3 (6:30 a.m. and 6:30 p.m. local solar equator-crossing time; ECT) and 6 (3:30 a.m. and 3:30 p.m. local solar ECT) hours prior to IASI overpass time (Figs. 14, S7 and 8 respectively) to further examine the impacts of precipitation on the day-night differences in dust. We think precipitation at IASI overpass time also provide useful information of concurrent meteorological conditions in the climatology so we prefer to keep it (Fig. 14). Overall, precipitation pattern at 6 hours prior to IASI overpass time looks similar to that of 3 hours prior but with stronger magnitudes. Day-night differences in precipitation shows little significance area except a few spots in the northern Sahara in JJA for all the above time steps. This indicate that wet deposition may not be playing any significant role in controlling the observed negative day-night differences in IASI DOD. Later, we also examined soil moisture, which is

influenced by both concurrent and previous precipitation and is more directly related to dust emissions. We found soil moisture also shows insignificant differences between daytime and nighttime (lines 759-762).

8. Please be advised (and mention in the text, and discuss especially in sections 3.3 and 3.4) a number of important information regarding the different data sets used. For both IASI and AERONET only cloud-free observations are possible, while (to be confirmed as I am not 100% certain) PM10 measurements occur under all meteo conditions. PM10 relates to the surface concentration, why DOD and CAOD are integrated along the whole column, making the comparison very difficult. Finally, IASI DOD contains only dust coarse mode, while AERONET coarse mode also contains other coarse particles (as sea salt – this is clearly stated in the manuscript), and PM10 contains all sorts of aerosols with size below 10µm – including for dust both fine and coarse particles (and other aerosol types).

We appreciate the advice. These are all issues we have acknowledged in the manuscript. In lines 205 and 210 we mentioned that level 2 and 1.5 AEROENT data are cloudscreened, and in lines 173-174 we mentioned IASI DOD is cloud screened. We also revised lines 388-390 to clarify the differences in IASI DOD and near surface  $PM_{10}$ concentrations. In sections 3.3-3.4, PM10 and IASI DOD (Fig. 6), AEROENT CAOD and IASI DOD (Fig. 7) are all collocated within the ± 30 minutes of IASI overpass time and within ~30 km radius of the LISA or AERONET stations.

9. It is important to discuss the fact that PM10 / DOD is highly variable in time, including between different years (linked to the also variable meteo conditions). Therefore comparing the seasonal / diurnal cycles of dust using different years of data requires being very careful about the conclusions. The paper lacks that and the information of the time span (and data holes) for all ground-based AERONET and LISA stations.

Thank you for the suggestion. we modify line 300 to clarify the temporal coverage of the LISA  $PM_{10}$  data. As we mentioned in our reply to the previous comment, now all station data are collocated with IASI to facilitate comparisons of seasonal cycles (Figs. 6-7), i.e., station data are sampled within ± 30 minutes of IASI overpass time while IASI pixels are sampled within a 30 km radius of the LISA or AERONET stations. While for diurnal cycles of AEROENT CAOD or LISA PM10, we used all the available station data (Fig. 8).

**Specific major comments:**

1. Lines 51-52: the radiative effects of dust also include the thermal infrared dust emissions, often forgotten by people working at short wavelengths, but that are also very important in the total radiative balance! Especially here the authors work with thermal IR observations so they should be aware and mention fully (i.e., not only absorption and scattering) the dust radiative effect at those wavelengths.

We modify lines 47–48 to reflect this suggestion: "The radiative effect of dust refers to its scattering and absorption of incoming shortwave and outgoing longwave radiation as well as thermal infrared emissions." We also added lines 105–106 to discuss the retrieval of

dust aerosols from IR band: "Additionally, coarse mode dust aerosols (CAOD e.g., radius  $> 1 \mu m$ ) are more sensitive to infrared (IR) radiation than visible due to their large particle size, so are preferentially retrieved in IR bands (Capelle et al., 2018)."

2. Figure 1: stations names are very difficult to read; maybe a separate map with the stations would be clearer? In addition, the difference between teal and royal blue is not that obvious.

We updated Fig. 1 to increase the font size of station names and modified font color. Only AERONET stations with both daytime and nighttime observations (at the same days) that are used to examine the day-night differences in CAOD as well as LISA stations are displayed. We have also included a clear map to display all the AEROENT stations used in this study in the supplementary materials (Fig. S1).

3. Figure 2: I see, for some sites (e.g. Dak, Mez, Sol, Kar, ...) more than others, a clear overestimation of IASI DOD (or underestimation of AERONET CAOD) for large DOD values. This should be discussed.

We revised our collocation method where AERONET CAOD is now sampled within  $\pm$  30 minutes of IASI overpass time while IASI pixels are sampled within a 30 km radius of AERONET stations. We also used the same IR/VIS ratio to convert IASI 10 µm DOD to 500 nm at each station. In the revised manuscript, we increased the number of solar sites used for the evaluation from 17 to 46, and lunar sites from 10 to 11. Using scatter diagrams to show relationship between IASI DOD and AERONET CAOD at each site is no longer possible, hence we show both Taylor diagrams and density plot (Figs. 2-3) to compare AEROENT CAOD and IASI DOD. Fig. 3 shows some overestimation of IASI DOD for small AEROENT CAOD and this is discussed in lines 343-345: "We also notice there are some overestimations of IASI DOD for small CAOD values (<0.5) in both daytime and nighttime records (mainly over coastal sites, such as, Mez, Eil, and Dak in daytime and Mig, Mez, and Bho at nighttime), which warrant future investigation."

4. Line 302 -> Could you be more specific than "close to" and "far from"? Is "far from" still in the area where a significant amount of dust is transported or do you mean so far that those areas usually don't see much dust? In the latter case, I would understand why the correlation would be worse. In the first case, I do not understand why the correlation would be lower: at sites where mostly transported dust is observed, the situation is much less variable than closer to sources, and therefore the comparisons should be better, I think.

Thank you for pointing this out. "Close" means either at or very near dust source regions but "far" refers to the sites along the path where dust is transported from source regions. We have modified lines 355-367 to clarify:" In general, IASI DOD at sites around dust source regions is better correlated with AERONET CAOD than sites around regions where dust is transported from source regions (e.g., the southern Sahel), and worsened in areas characterized by complex terrains and pollutants from either biomass burning, industrial emissions or coastal sediments (e.g., Eilat)"

5. Figure 3: I see here (opposite to Fig 2) some underestimation of low AOD by IASI (or overestimation by AERONET lunar), again for example in Dak, Mez, Sha, Kan. This should be

The revised figure (Fig. 3), where AERONET and IASI are better collocated, does not show such obvious underestimations of IASI DOD. The correlations between IASI DOD and AERONET CAOD are generally lower for lunar sites than solar sites and lunar sites also show higher centered RMSE (Fig. 2). The quality of the level 1.5 lunar CAOD (cloud-screened but not quality-controlled) and smaller sample size may partly contribute to the discrepancies between IASI and AERONET.

6. Figures S1 and S2: one should not compare seasonal cycles over different years. Those cycles may be different from year to year, linked to local meteorological variability. I am aware that this makes the process more difficult, as some stations do not provide data for parts of some years, or do not span the same time frame, while IASI observations occur (almost) every day. But at least it should be mentioned that (or tested if) some differences might be due to sampling different years / months with the data that are compared. A possibility is, for these comparisons, to remove the IASI data when there is no data for the corresponding ground-based station.

Thank you so much for raising this point. We want to clarify that the seasonal cycles of AERONET and IASI are compared over same years for each site, but years largely varies from site to site due to the limited amount of datapoints available at each AERONET sites. In the revised manuscript, we have replaced Figs. S1 and S2 with the seasonal cycle of dust in Fig. 7, where AREONET CAOD and IASI DOD are collocated, i.e., CAOD are sampled within  $\pm$  30 minutes of IASI overpass time, while IASI DOD pixels are averaged within a 30 km radius of the AERONET stations.

7. Lines 351-372: why differences are discussed while they are not significant? This means just discussing noise. Only the significant difference should be mentioned, i.e. only the LISA Cin data. For the comparison of the seasonal cycles between IASI and the stations, the next section 3.3 is much more precise and readable than the stars and dots in Figure 4.

Thanks for the comment. Even though majority of the stations are insignificant, the daynight differences of some sites show similarity with IASI, and we want to illustrate these similarities. Many are insignificant probably because of the smaller sample sizes of AERONET lunar measurements. We have revised lines 384-394 to focus on sites with significant day-night differences in CAOD.

8. Lines 394-395: in addition to what the authors mention, I think that the very narrow swath of CALIOP also plays a role in the differences with IASI. With a repeat time of 16 days, a place is observed only twice per month (missing many events) and not the same day for afternoon and night observations (in addition occurring with a clearly different quality), all of it making the differences between CALIOP early afternoon and night very dubious, to my opinion.

Thanks for the comment. We totally agree and briefly mentioned the narrow swath of CALIOP in lines 96-97. We revised these lines to improve clarity: "A narrow horizontal swath of 5 km and a 16-day repeat cycle, which means there is only one daily observation (afternoon or night) at a specific location thus no day-night differences of DOD can be retrieved at the daily timescale." We also added the influence of narrow swath of CALIOP to lines 418-421:" Moreover, because of the narrow swath width of CALIOP with a sampling rate of twice per month (repeat cycle of 16 days), the afternoon and night observations are not at the same day, which also influence the day-night differences in CALIOP DOD."

9. Line 398: only the old papers are mentioned here, the more recent dust layer height validation paper of Kylling should also be mentioned here, with its conclusions. (I see that it is mentioned lines 407-408 but only for its methodology, not the resulting bias of IASI versus CALIOP)

Manuscript has been revised to include relevant parts of the results by Kylling et al. (2018) in line 432. The bias of IASI versus CALIOP by Kylling et al., (2018) is also added in lines 432-435.

10. Figure 6: please use the same vertical scale for the different stations to allow comparisons. Also, I am confused now looking at those plots with errors bars: for example Ban DJF -> all day are significantly higher than night in Fig 6 but the difference is not significant in Fig 5. Again, I think the station differences should be removed from Fig 5 where in any case they are very difficult to see and reported non-significant.

Fig. 6 has been revised to use the same vertical scale for the  $PM_{10}$  and CAOD plots. And daytime PM10 at Ban is significantly higher than nighttime at the 90% confidence level (Fig. 4). We prefer to keep the station data in Fig. 4 as station data provide additional spatial information of PM10 and CAOD to be compared with IASI DOD for daytime, nighttime, and day-night differences.

11. Lines 420-422: the differences at MBour between IASI and PM10 can't be just because sea salt is included in the PM10. That explanation holds for a larger PM10 during the winter at that station but not for a lower PM10 at that station during spring and summer. Another explanation for the difference between IASI and PM10 would be that MBour is further from the sources, hence the dust is higher in the atmosphere and not recorded in surface PM10 while recorded in IASI column DOD. Or possibly that there is more wind, actively dispersing the dust therefore making the average PM10 smaller. Maybe those hypotheses could be checked.

Thanks for the suggestions. The manuscript is revised accordingly to incorporate these comments in lines 459-452: "This disparity could partially be due to the fact that Mbo site is located along the transport path of the boreal JJA dust plumes, but further from the major dust sources of North Africa, thus dust aerosols are likely mixed to higher altitudes which may be sampled differently between the near nadir viewing IASI instrument and the surface measurements, and differently between day and night.".

12. Figure 7: again please use the same vertical scale for all sub-figures to allow comparisons. In addition, it would be much more clear if IASI or AERONET was converted (using a once-decided standard way all along the paper) and then both plotted in the same plot for much easier comparison. Again the temporal range for the different stations, which does not match the full range, makes the comparisons more difficult. See also major comment 6.

Figure. 7 has been updated that all sub-figures now use the same vertical scale. . We collocated IASI DOD with AERONET CAOD by sampling CAOD within the  $\pm$  30 minutes of IASI overpass time and IASI DOD pixels within a 30 km radius of the AERONET stations. We have also scaled IASI 10  $\mu m$  DOD to visible equivalent DOD at 500 nm using a uniform IR/VIS ratio for all the stations.

13. Figure 8: same comment about the vertical scale; in addition I think the text lacks some discussion about the differences e.eg. between MBour and Dakar which are very close and yet very different. Again, I think the key is in the fact that the observations are totally different and should actually not be compared without explanations. The sentence lines 472-473 should contain some discussions and not just a note of the differences or similarities depending on the site.

We thank the reviewer for the helpful comment. We decided to keep the different vertical scales for individual stations as the magnitudes of PM10 and CAOD vary greatly among the stations. For some sites with smaller magnitudes the diurnal cycles cannot be clearly displayed if the same vertical scale is applied. We totally agree that PM10 and CAOD reveal different aspects of dust distribution in the atmosphere and not comparable. While CAOD reveals column dust loading, PM10 concentrations reveal near-surface abundance of dust. We show them together to better understand the diurnal variations in dust as both datasets have high-temporal resolutions diurnal records. The reasons for the differences between M'Bour and Dakar are discussed in lines 455-459. Moreover, Dakar and M'Bour show different diurnal cycles as PM10 and CAOD reveal different aspects of dust distribution.

14. Line 522: please clarify (again) what is meant here by daytime and nighttime wind speed; it looks as if it is an average along the whole day (or night);

Figure 11 shows wind speeds and direction corresponding to IASI overpass times i.e., 9:30 a.m. (9:30 p.m.) local solar ECT for daytime (nighttime). We have revised the entire manuscript to clarify this.

15. Figure 11: I can't see the wind vectors properly (figures are too small)

Figure 11 has been updated to better display wind vectors.

16. Line 574-576: I do not understand. Why here highlight the lack of match between maximum PM10 and minimum wind speed? We should not expect such a match, or should we? And what other factors would you suggest contribute to evening increase in PM10 than wind speed? Please elaborate.

We removed the above lines during our revision.

17. Paragraph lines 577 to 585 is largely unclear to me. The morning peaks and afternoon minima of wind speed do not occur at all stations. The comparisons are much confusing. In addition, again I would not expect a full match between surface wind speed and total column AOD / DOD (both because winds do not always lead to emissions, and because surface is different from column observation).

Thanks for the comment. We revised lines 586 to 605 to better compare the diurnal cycle of surface winds with that of CAOD and PM10. We also did not expect a full match between surface wind speed and CAOD for all sites. That's why in the discussion we highlighted stations showing consistency between surface winds peak/minimum with CAOD peak/minimum in different regions in the above lines. For instance, the morning peaks in wind speed occur over only sites in North Africa whereas afternoon peaks are observed in the Middle East sites. These times seem to correspond with maximum CAOD over North Africa in the morning hours and over Middle East (especially in Mig, Sha, and Mez) in the late afternoon hours. We agree that winds do not aways lead to emissions. This is particularly true over sites far away from dust sources and characterized by high soil moisture and vegetative cover i.e., the Sahel, the Guinea Coast, and the central to the southern Indian Subcontinent. However, higher wind speed over the Sahara in DJF will lead to higher dust concentrations over the Sahel (Kof and Ilo).

18. Lines 631-633: please recall that AERONET does not provide observations under cloudy sky, so also not when there is precipitation. Therefore, the AERONET observations peaking at the times of the precipitation peaks simply most probably do not occur on the same days. This is, I guess, easy to check in the data.

Thank you for pointing this out. We revised lines 635-654 to better discuss the diurnal time series of AERONET CAOD and IMERG precipitation. In lines 647-649 we highlighted that AERONET data are cloud screened: "Also note that since AEROENT CAOD data are cloud screened, few records are available during precipitation-prone hours. Thus, the scavenging effect of precipitation on dust may not be fully illustrated on the AERONET CAOD time series. "

19. Lines 636-637: how can low winds enhance dust emissions? Or maybe low altitude winds is meant?

Low altitude winds are intended. We revised line 657 to avoid confusion.

20. Lines 640-642: "of solar radiation reaching the surface" -> technically, if the radiation has reached the surface it was not absorbed, so I would remove "reaching the surface"; in addition, the thermal emission effects is again lacking from the dust-radiation interactions. I think the impact on the PBL height is not so straightforward: it will depend on the local conditions, because the absorption and scattering of solar light tends to reduce the heat, while emission of thermal light tends to increase the heat.

Thanks for the comment. We removed "reaching the surface" in the sentence. As the reviewer pointed out, while absorbing aerosols, such as dust, reduce the solar fluxes to the surface, they can increase downward longwave radiation at the surface due to the emission of infrared fluxes. The overall effect depends on the optical properties of dust, but usually a reduction of net surface fluxes, thus reducing the energy to generate a deep PBL. We therefore revised lines 661-664 to avoid confusion: "High concentrations of absorbing dust aerosols within the boundary layer can enhance the absorption and scattering of significant amount of solar radiation, decreasing the net radiation at the surface, which can reduce the sensible heat fluxes needed to drive the PBL evolution leading to a much shallow PBL height (PBLH; Li et al., 2017)".

21. Line 715: events occurring before the passage of IASI would not be missed if they are large enough, as the dust will remain in suspension for at least a few hours, possibly up to days. Events occurring after the IASI observation then also occur before the next IASI observation and can be observed if they last long enough. They would however be "displaced" depending on the winds and the time between the event and the observation. I would rephrase and nuance that part of the discussion.

Thanks for the comment. We have revised lines 737-740 to include this suggestion: "some small dust events before and after the passage of IASI (9:30 a.m. and 9:30 p.m. local solar ECT) could be missed. For those large events that occur a few hours before or after IASI observations, while IASI may still be able to capture them, the location of dust plumes is often shifted from their original locations depending on the direction of the prevailing winds"

22. Line 726: please elaborate. Do you mean a negligible impact of soil moisture difference between the different overpass times? This requires more than just a sentence. Indeed, if the soil moisture remains the same for morning and evening, then its impact on dust emission differences between morning and evening would be negligible. However, I would be very surprised if indeed the soil moisture does not show a diurnal cycle, depending on precipitation, surface type, temperature, solar heating, ..... Maybe it is just that ERA-5 is not able to reproduce these cycles?

Soil moisture does show some differences between the different overpass times of IASI, however, such day-night difference is small and insignificant over large spatial scales. Besides, the impact of soil moisture is mostly felt at the southern Sahel, the Guinea Coast, and the central to southern parts of the Indian Peninsula which are unfortunately not major dust sources, hence cannot significantly influence the day-night differences in DOD. We have modified lines 765-767 to further clarify: "However, our examination of soil moisture from ERA5 showed that the difference in soil moisture between IASI daytime (9:30 a.m. local solar ECT) and nighttime (9:30 p.m. local solar ECT) overpasses is small and insignificant, indicating a likely negligible impact on the day-night differences in DOD."

23. The discussion focuses mostly on IASI shortcomings, and should also contain the whole "problem" of comparing different data from different instruments observing different

quantities at different times under different meteo conditions, and the reasons of uncertainties in the other data sets compared in this work.

We have added a paragraph that discusses the uncertainties associated with station and the potential problems in comparing different products from different observational platforms (lines 747-760).

24. Conclusions need to be modified according to other changes in the manuscript

Our revision did not change the major findings of this work but a few minor details. We have made updates accordingly in the Conclusions section.

**Minor comments**

1. Abstract, lines 28-30 : this is confusing... only the morning hour time frames are provided, then for "late afternoon" no time frame is given; and the sentence is a bit long, making it difficult to read.

We have shortened the sentence to make it more readable (lines 27-30).

2. The abstract is a bit too long, I think

We have shortened the abstract to one page.

3. Line 40: "It is produced mainly by wind erosion ..." -> why "mainly"? Is there another mechanism?

We have removed "mainly" to avoid confusion.

4. Line 50: "the effect of which" makes it unclear to me if it refers to only the latter (dust - cloud interaction) or both (dust – radiation and dust-cloud interactions)

The indirect interaction of dust with radiation through clouds is hard to quantify and therefore serves as the largest sources of uncertainties in modeling aerosols effects. We intended to convey this message to the reader, but your concern is greatly appreciated. We have revised the sentence accordingly.

5. Line 51: could you add some more recent references about dust-induced uncertainties in modelling aerosol impacts on climate change ?

A few more most recent papers have been added (line 47)

6. Line 53: 4 of the citations (over 7) about the effects of dust on Monsoon are of the team writing the current publication, does it represent the full literature "statistics" (at first glance it feels like a way to increase the team's papers reference numbers; I would think only one reference is enough especially since the titles are relatively similar)

Thank you for raising this point. The co-author has been working on Indian-dust monsoon interactions for many years and all the papers touched different aspects of monsoon-dust interactions. However, to avoid any speculation, we removed two references.

7. Line 132, line 140; line 141: maybe add some more recent references for the most important dust sources in each part of the dust belt?

Thanks for this suggestion, the following references are added to the Saharan dust sources: Schepanski et al., 2007; Ginoux et al., 2012; Yu et al., 2018, the Middle East dust sources: (Ginoux et al., 2012; Yu et al., 2018), and East Asian dust source: (Zhang et al., 2003; Ginoux et al., 2012)

8. Line 152: the section title needs to make clear which IASI data is used

The title of section 2.2.1 is revised to "LMD IASI".

9. Lines 106 and 154: I suggest using the original IASI papers when describing the instrument, not the papers of one specific user of the data, which also cites the original papers when describing the instrument

The original references have been added to lines 101, 104, and 155.

10. Line 155: the spectral resolution of 0.5cm-1 is after apodization, not the original resolution of the instrument (but indeed that of the data most commonly used) -> just add "after apodization" in the sentence

We appreciate the correction. "after apodization" has been added to lines 157.

11. Line 156: why only "onboard Metop-A"? I understand that here only data from Metop-A is used, but the general description is the same for all IASI instruments onboard the 3 Metop satellites

We used Metop-A because it has the longest records for our study. Observation started since July 2007 up to 2021.

12. Line 156: "at an angle of 48.5°" -> this is the maximum value, not a constant value.

We have modified the sentence to incorporate this in lines 158 "IASI observes Earth at an angle of up to 48.5° perpendicular to both sides of the satellite track".

13. Line 157: the IASI swath is about 2200km and one IASI instrument leads to almost global coverage in 12 hours (not really global, some bands are missing between orbits at low latitudes)

Thanks for pointing this out. We have modified the sentence to: "This corresponds to a swath width of  $\sim$ 2,200 km leading to an approximate global coverage in 12 hours" in line 159.

14. Line 158: "the satellite has a local equatorial crossing time" -> local solar equatorial crossing time

Suggestion is much appreciated. Corrections have been made.

15. Line 158: the availability mentioned here is indeed for IASI / Metop-A but it should be made clear that the other instruments have different time frames and the temporal series of consistent data is continued

This is a great suggestion. We have revised text accordingly in lines 162-164 "MetOP-B was launched in September 2012 and has been operational since February 2013, while MetOP-C was launched at the end of 2017 and has been providing data since 2018. With MetOP-A coming to an end, MetOP-B and -C will continue providing data with the three instruments expected to provide measurements up to a total of 15 years."

16. Line 160: missing some refs about the use of IASI for atmospheric composition

We have added references on the application of IASI for atmospheric composition retrievals in lines 165-166: "Due to its wide spectrum in longwave range and fine spectral resolution, IASI is widely used to retrieve atmospheric compositions (Clerbaux et al., 2009; Bauduin et al., 2016)".

17. Line 163: Météorologie (check the accents)

**Corrected.**

18. Line 164: "with in situ observations" – please do not mix AERONET (ground-based remote sensing) with "in situ" (real local observations, such as PM10 if really obtained from local aerosols as the LISA measurements)

**Corrections have been made.**

- 19. Line 175: "CALIOP is a near-nadir": no, CALIOP is purely nadir Corrected.
- 20. Lines 176-177: I don't see how the 16 days repeat cycle "makes" the field of view what it is ... Please rephrase

The text has been rephrased to: "a beam diameter of 70 m at the Earth's surface corresponding to a 16-day repetition cycle with an instantaneous field of view approximately 300 m and 70 m" in lines 188-189.

21. Line 189: why provide the field of view only for the lunar observations?

Both observations have almost similar field of view. Solar and lunar field of views are respectively 1.20 and 1.29. Text is modified accordingly.

22. Line 191: formulation is weird, what means >+- ? Uncertainty is, by nature, a box around the value, so I would omit the +-. As for the > signs: do you mean that 0.01 is the maximum uncertainty for wavelengths larger than 0.44¬μm, while 0.021 is the minimum uncertainty for shorter wavelengths? This is very unclear, I do not know how to interpret minimum or maximum uncertainties in that context. Please rephrase.

Thank you for raising this point. You are right to point out the confusing nature of the statement in the text. The text is modified to avoid confusions in line 203.

23. Line 292 and following: please explain why you used SDA (I agree with the choice, but the explanation is missing)

We used SDA because in the SDA, AOD is well partitioned between fine and coarse mode aerosols using the AOD spectral properties. We mentioned this in line 205-206: "We use version 3 level 2 (cloud screened and quality assured) Spectral Deconvolution Algorithm (SDA; O'Neill et al., 2003) retrieval of the coarse mode AOD (CAOD; Eck et al., 2010) around 500 nm to approximate DOD and compare with IASI DOD."

24. Line 198: How do you remove missing values?

We removed the line.

25. Line 201: Six sites are supposedly blue dots in Fig. 1 - I only see 5 of them

That was an oversight on our part. It appears a wrong figure was used. The whole figure has been revised with new sites clearly and boldly shown.

26. Line 226: Shifted to local time -> local solar time

The whole sentenced has been revised to take care of the mistake.

27. Title 2.3 -> should start with "validation of"

We changed the section title to: "Validation of IASI DOD against AERONET station observations"

28. Figure 4 caption vs text line 320: please also mention in the text that a mean scaling factor is used

**Corrected.**

**Technical corrections**

Abstract, line 7: probably missing the word "data" after the end of the parenthesis

We are not sure about where "data" fits. However, lines 7-8 are now modified to: "Utilizing the well-calibrated, high spectral resolution, and equal-quality of daytime and nighttime (9:30 a.m. and 9:30 p.m. local solar equator-crossing time) products of the Infrared Atmospheric Sounder Interferometer (IASI) from Laboratoire de Météorologie Dynamique (LMD),..."

Abstract, line 9 : both ... show (without the s)

Corrected.

Abstract, line 18, I think it misses "night time" before "dust emissions"

Corrected.

Lines 39-40: "it is produced [...] their uplift" -> choose singular ("the dust" or plural "dust aerosols"; I would go for plural)

Corrected.

Line 41: "uplift [...] occur" -> occurs

Corrected.

Line 254 "at" should be without capital letter

Thanks for catching that, corrections have been made.

Line 264: AERONET is mis-spelled

Corrected.

Line 319: add for IASI that it is at 10µm (to avoid any possible confusion)

10 µm is added to avoid confusion.

Line 349 "a uniform scaling factor [...] is used" (instead of "are used")

Line has been corrected.

Line 358 AERONET is mis-spelled

Corrected.

Line 393: "in general" or "generally"

**Corrected.**

Line 444: "five to 15" -> "5 to 15"

**Corrected.**

Line 459: Maybe "at" is missing at the beginning of the sentence?

Yes, but that was covered by the figure. Corrections have been made.

Line 474: two time steps (no -)

**Corrected.**

Line 521 Figure 11 showS

Corrected.

Line 566: weird formulation: what are dust activates?

This was supposed to be "activities" not "activates". That has been replaced by "Aerosols".

Line 660: remove "the" before day-night

Corrections have been made.

Line 719: I guess "orbital gaps"

Good catch, "gabs" is replaced by "gaps".

References:

Carboni, E., Smith, A., Grainger, R., Dudhia, A., Thomas, G., Peters, D., Walker, J., and Siddans, R.: Satellite remote sensing of volcanic plume from Infrared Atmospheric Sounding Interferometer (IASI): results for recent eruptions., in: EGU General Assembly Conference Abstracts, EGU2013-11865, 2013.

Chalon, G., Cayla, F., and Diebel, D.: IASI- An advanced sounder for operational meteorology, in: IAF, International Astronautical Congress, 52 nd, Toulouse, France, 2001.

Clarisse, L., Clerbaux, C., Franco, B., Hadji-Lazaro, J., Whitburn, S., Kopp, A. K., Hurtmans, D., and Coheur, P.-F.: A Decadal Data Set of Global Atmospheric Dust Retrieved From IASI Satellite Measurements, J. Geophys. Res. Atmospheres, 124, 1618–1647, https://doi.org/10.1029/2018JD029701, 2019.

Klüser, L., Vandenbussche, S., Capelle V., Clarisse, L., Kalashnikova, O., Garay, M. J., Popp, T.: IASI dust algorithm inter-comparison within ESA's Climate Change Initiative, 3rd AeroSAT

Meeting, 2015. (https://aero-sat.org/sites/wdc.dlr.de.esa-aerosol-cci/files/sites/wdc.dlr.de.esa-aerosol-cci/meeting3/AS3\_new\_IASI.pdf; last access Jan. 2023).

---

## Author Comment (AC2)

Jacob Zora-Oni Tindan

Department of Meteorology

Pennsylvania State University

530 Walker Building

University Park

State College, PA, USA.

The Editor

Atmospheric Chemistry and Physics

Copernicus Publications.

February 7, 2023.

Dear Editor,

We have submitted a revised paper entitled "Understanding day-night differences in dust aerosols over the dust belt of North Africa, the Middle East, and Asia" by J. Z. Tindan, Q. Jin, and B. Pu for consideration for *Atmospheric Chemistry and Physics*. The helpful and constructive comments from two the anonymous reviewers are deeply appreciated. Our replies to each reviewer's comments are attached. We mainly made the following changes:

- Following the comments from Reviewer #2, more precise IASI overpass time from level 2 data is used in our analysis. Correspondingly, we revised our method to collocate IASI DOD and AERONET CAOD for data validation. Now AERONET station data are sampled within ± 30 minutes of IASI overpass time, while IASI pixels are averaged within a 30 km radius of the AERONET stations. Reanalysis variables are resampled at each grid point based on IASI overpass time, so that meteorological variables are at the same time as IASI retrievals to facilitate comparison. All the related figures and text are updated.
- We refined our selection of AERONET sites. Now 46 solar sites and 11 lunar sites are used to evaluate IASI daytime and nighttime DOD, and nine stations are used to examine the day-night differences in CAOD.
- We now apply a uniform infrared to visible bands (IR/VIS) conversion ratio, 0.6, to convert IASI 10 μm DOD to its 500nm equivalent for both data validation and analysis, following the suggestion of Reviewer #2.
- We have performed additional analysis of wind speed and precipitation at 3 and 6 hours before IASI overpass time to determine whether these variables at previous times have impact on the distribution of DOD at IASI overpass time.

- Following the comments from Reviewer #2, we emphasize that we used IASI DOD and dust layer height retrieved by Laboratoire de Météorologie Dynamique (LMD) in this study. The reasons to use LMD IASI products are also discussed.
- following the suggestion of Reviewer #2, we added some discussion about the uncertainties associated with station data and the potential problems in comparing different products from different observational platforms.

We also made some minor edits to improve the clarity of the paper.

Thank you again for your time and consideration! We are looking forward to hearing from you.

Sincerely,

Jacob Zora-Oni Tindan, Qinjian Jin, and Bing Pu

---

## Author Response (AR2)

Jacob Z. Tindan

Department of Meteorology

Pennsylvania State University

530 Walker Building

University Park

State College, PA, USA.

The Editor

Atmospheric Chemistry and Physics

Copernicus Publications.

March 31, 2023.

Dear Editor,

We have submitted a revised paper entitled "Understanding day-night differences in dust aerosols over the dust belt of North Africa, the Middle East, and Asia" by J. Z. Tindan, Q. Jin, and B. Pu for consideration for publication in *Atmospheric Chemistry and Physics*. The helpful and constructive comments from the two anonymous reviewers are deeply appreciated.

Following the comments from Report #1, we have checked the spelling of AERONET across the entire document and made the necessary corrections. We have also made some minor edits to both text and figure captions and labels to improve clarity of the paper.

Thank you again for your time and consideration! We are looking forward to hearing from you.

Sincerely,

Jacob Z. Tindan, Qinjian Jin, and Bing Pu

**Responses to Reviewers' Comments**

We would like to express our sincere gratitude and appreciations once again to the reviewers for their valuable and constructive suggestions. The comments and concerns of reviewer #2 in report #1 is addressed below. Line numbers refer to the lines in the manuscript with tracked changes.

**Comments in report #1:**

please check the spelling of AERONET all along the paper

Thank you very much for catching the errors. We have corrected these spelling errors across the entire document.

In section 3.1, please avoid using bias of -100%, for example. Either mention a negative bias of 100% or an underestimation by 100% (without sign, being comprised in the "negative" or "under" words)

We appreciate this input. Section 3.1 lines 378-380 have been corrected as "and a large negative bias of more than 100% (see Table 2 and Fig. 2a). Similar large negative biases are also observed around other coastal sites over North Africa (e.g., Iza, Lag, and Tei) possibly due to the mixing of sea salt with dust aerosols and the complicated land surface conditions in the area leading to difficulties in DOD retrieval (Capelle et al., 2014, 2018). Similarly, nighttime DOD is also underestimated at Tei and Iza by more than 200%"